# Fast evolutionary turnover and overlapping variances of sex-biased gene expression patterns defy a simple binary sex classification of somatic tissues

Chen Xie[1,2], Sven Künzel[1], Diethard Tautz[1]*

[1]Max Planck Institute for Evolutionary Biology, Plön, Germany; [2]Biomedical Pioneering Innovation Center, Peking University, Beijing, China

## eLife Assessment

This study presents data on sex differences in gene expression across organs of four mice taxa. The authors have generated a unique and **convincing** dataset that fills a gap left by previous studies. They claim that sex-biased expression in the soma can overlap between genetic males and females, and that the relevant patterns both turn over quickly over short evolutionary times and do so faster in somatic than gonadal tissues. These conclusions could largely have been predicted by extrapolating from previous findings in the field, but nevertheless demonstrating them directly is a **fundamental** advance.

[Editorial note: The work was originally assessed by colleagues who are active in the field of evolution of sex differences or in areas adjacent to this field (see initial assessment at https://doi.org/10.7554/eLife.99602.2). The appeals process involved consultation with experts working in other areas of evolutionary biology. The above assessment synthesises the opinions of both sets of reviewers.]

*For correspondence:
tautz@evolbio.mpg.de

**Abstract** Sexual dimorphism in phenotypes is largely driven by genes with sex-biased expression, spanning from key regulators to numerous organ-specific effectors. Current understanding is limited regarding the evolutionary dynamics of these genes in somatic tissues that generate the adult phenotype versus gonadal organs that are required for reproduction. Here, we investigate sex-biased gene expression and micro-evolutionary patterns of these genes in populations of subspecies and species of wild mice (genus *Mus*) that were raised under controlled conditions. We find a faster evolutionary turnover of sex-biased gene expression in somatic tissues, but not in the gonads, when compared to the turnover of non-sex-biased genes. We introduce a sex-biased gene expression index (SBI) to quantify individual variances. We find a range from binary to overlapping SBI patterns across individuals. SBI values do not correlate between organs of the same individuals, thus supporting a mosaic model of somatic sex determination. Comparison with data from humans shows mostly fewer sex-biased genes compared to mice and strongly overlapping SBI distributions between the somatic organs of the sexes. We conclude that adult individuals are composed of a mosaic spectrum of sex characteristics in their somatic tissues that should not be cumulated into a simple binary classification.

## Introduction

The usual narrative for sex-related differences between individuals is based on binary gametic sex (*Goymann et al., 2023*). Such a binary distinction is a hallmark of most sexually reproducing species,

and it is therefore often considered a cornerstone of evolutionary conservation. However, Darwin has already pointed out that the phenotypes of the sexes can evolve fast when they are subject to sexual selection (*Darwin, 1871*). The different interests of males and females lead to sexual conflict, characterized by opposing evolutionary constraints on the genes that mediate sex differences. Sexual conflict can have many forms but can be broadly categorized into interlocus and intralocus sexual conflict, with distinct implications for evolutionary biology and species fitness (*Schenkel et al., 2018*). Each is also expected to have an effect on the evolution of the underlying genes and their regulation (*Ingleby et al., 2015*; *Mank, 2017a*; *Tosto et al., 2023*).

The sexual phenotypes of adult individuals are generated by hundreds to thousands of genes with sex-biased expression (*Mank and Rideout, 2021*; *Parsch and Ellegren, 2013*; *Tosto et al., 2023*). The combination of variants of these genes is expected to determine the range of somatic individual phenotypes of males and females (*Tosto et al., 2023*; *van der Bijl and Mank, 2021*). For example, human height is partially determined by genes with sex-biased expression, resulting in overall smaller females than males (*Naqvi et al., 2019*). However, height distributions of males and females are highly overlapping, such that knowledge of the height of a given individual would not be sufficient to classify it as either female or male. In fact, such overlapping distributions are characteristic of somatic sex-related phenotypic characters (*Maney, 2016*).

The studies of morphological sex differentiation of the human brain have revealed mosaic patterns where different brain parts of the same individual can show more female or more male characteristics (*Joel et al., 2015*). Hence, a binary narrative that is based on gametic sex is not applicable at this somatic level and could be replaced by relative degrees of maleness versus femaleness (*Joel et al., 2020*). Different organs are known to be shaped by different sets of sex-biased genes (*Naqvi et al., 2019*; *Rodríguez-Montes et al., 2023*), making the degrees of somatic sex-specific variation between individuals even larger.

Sex-biased gene expression is known to evolve fast, supporting the notions of sexual conflict theory (*Tosto et al., 2023*). However, the evolutionary comparisons of sex-biased gene expression have so far focused on comparisons between relatively distant species and relatively few individuals per species only (*Naqvi et al., 2019*; *Rodríguez-Montes et al., 2023*), making it difficult to gauge the true evolutionary rates. A dedicated study exploring the micro-evolutionary context of sex-biased gene expression in animals is missing so far.

Here, we use the house mouse radiation to address the evolution of sex-biased genes and their expression. The system is an excellent model to study micro-evolutionary turnover patterns at a shallow evolutionary population level, as well as between subspecies and species (*Harr et al., 2016*; *Zhang et al., 2021*). For our study, we use outbred individuals derived from natural populations from two *Mus musculus* subspecies, *M. m. domesticus* (DOM) and *M. m. musculus* (MUS), as well as two sister species, *M. spretus* (SPR) and *M. spicilegus* (SPI). To avoid confusion with species status, we designate all four with the neutral term 'taxon' and use only the three-letter abbreviations. The taxa cover an evolutionary distance of about 2 million years (*Figure 1*). Given the short reproduction cycles and the fast molecular evolution in rodents, this time corresponds, in terms of molecular divergence, approximately to the equivalent of the evolutionary split between gibbons and humans. Hence, the system allows for studying the continuum from micro-evolutionary to macro-evolutionary patterns. We use individuals derived from natural populations but kept under common environmental conditions and in an outbreeding regime to retain the natural variation (*Harr et al., 2016*).

Most previous studies on sex-biased genes have given special emphasis to the evolution of gene expression in gonadal tissues even when somatic tissues were co-sampled, since the gonads harbor the largest numbers of sex-biased genes (*Dean et al., 2017*; *Harrison et al., 2015*; *Pointer et al., 2013*; *Todd et al., 2018*). In contrast, our study is mostly focused on somatic tissues that occur in both sexes, but we complement it with comparisons to gonadal tissues that are specific for each sex. Since the gonadal tissues are composed of different cell types for each sex, a comparison of expression patterns is always a composite between gene expression and cell-type composition. This is different for somatic organs, where one can expect that the cell types are much more similar between the sexes.

It has recently been shown that the major onset of sex-biased gene expression is at the developmental transition to sexual maturation (*Rodríguez-Montes et al., 2023*), and we are therefore focusing on sexually mature stages. We use sets of 18 individuals from each taxon, which allows us

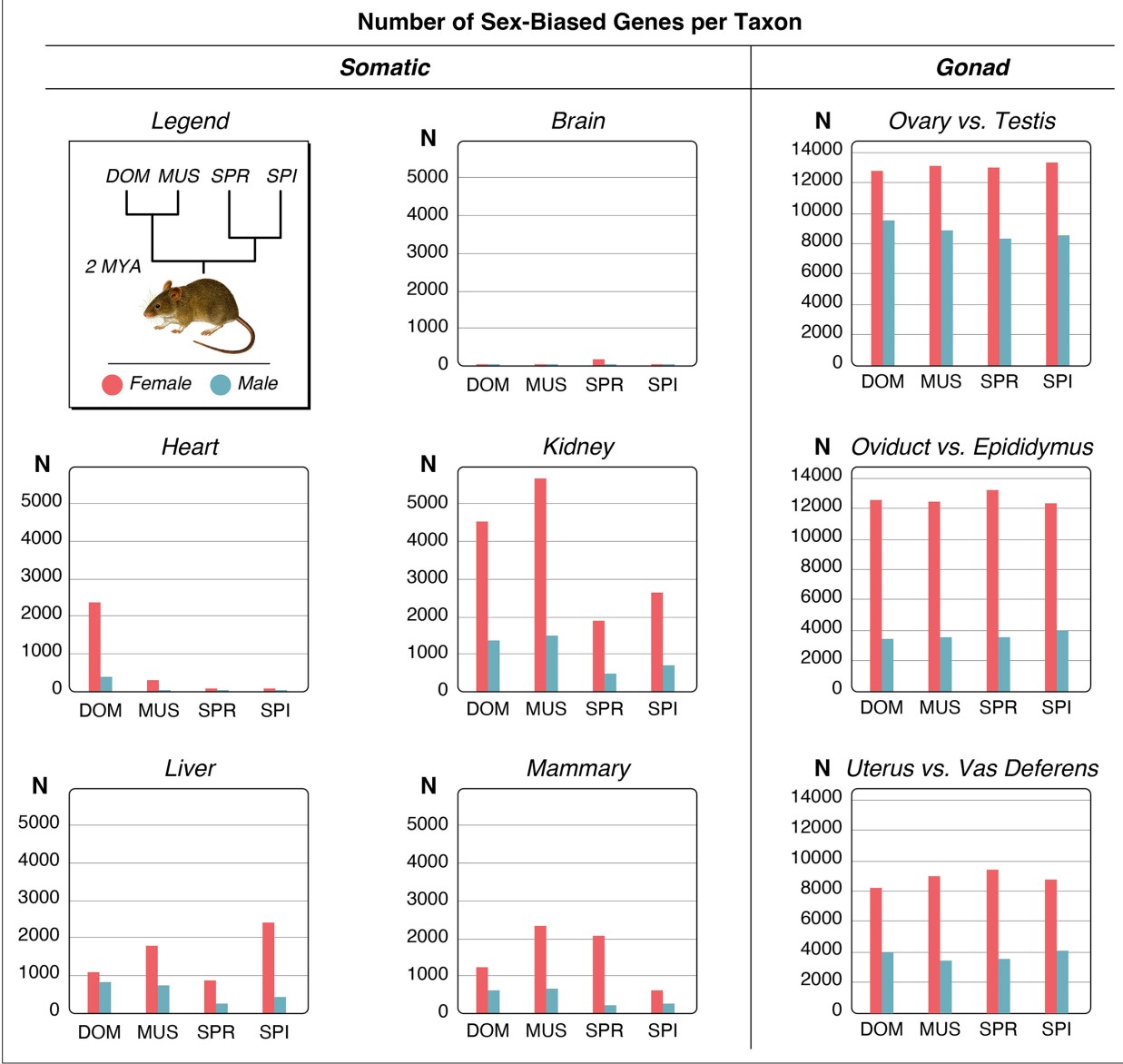

**Figure 1.** Relationship of the mouse taxa and numbers of sex-biased genes for each organ comparison. The top left of the plot shows the phylogenetic relationships of the taxa in the study. The other plots show the numbers of sex-biased genes as bar plots, female-biased in red, male-biased in green. DOM = *M. m. domesticus*, MUS = *M. m. musculus*, SPR = *M. spretus*, SPI = *M. spicilegus*. The Y-axes show the number of genes. Note that there are two different Y-axis scales for the somatic organs and the gonadal organs. All numbers are provided in *Figure 1—source data 1*, full data are provided in Supplementary Data D1.

The online version of this article includes the following source data for figure 1:

**Source data 1.** Numbers of sex-biased genes in mouse organs.

**Source data 2.** List of mouse samples included in the study.

**Source data 3.** Histograms of distributions of female/male sex-bias ratios for all mouse and human organ datasets.

**Source data 4.** Scatterplots of F/M ratio values for all genes that show a significant sex-bias change in the respective tissue and mouse taxon pair comparison.

**Source data 5.** Numbers of shared genes between organs for each of the taxa.

to compare within-group variation with divergence between groups, a contrast that has been lacking so far in most studies on sex-biased gene expression, except for humans (*Khodursky et al., 2022*).

We find thousands of sex-biased genes in the somatic organs, but with marked differences in numbers between the organs in the different evolutionary lineages. As it was previously found, most

sex-biased genes are organ-specific (*Naqvi et al., 2019*; *Rodríguez-Montes et al., 2023*), and we show here that they are expressed in organ-specific modules. Only a small percentage of genes shows conserved sex-biased expression patterns among all lineages, and the variances of expression between individuals are higher than for non-sex-biased genes.

To represent the individual variation in expression of sex-biased genes, we have developed a sex-biased gene expression index (SBI) that combines the sex-biased gene expression for each individual to reflect its overall femaleness or maleness in each organ. In the somatic organs, we find variable patterns of distinction and overlap between the sexes, while there is a strict binary pattern for the gonadal comparisons. We apply the SBI also to human data from the GTEx resource (*Aguet et al., 2020*) and find overall much fewer sex-biased genes when applying the same criteria as for the mouse data, and stronger overlaps between the sexes than in mice. The SBI can also be used to assess mosaicism of sex-biased gene expression between individuals analogous to brain sex-specific brain structures (*Joel et al., 2015*). We show that individuals are characterized by different degrees of mosaicism in different organs.

## Results

Gene expression data were collected from taxa and organs depicted in *Figure 1*. Nine age-matched adult females and males each were chosen as biological replicates among the outbred individuals from each of the four taxa, and the same set of organs was retrieved for each of them for RNA-Seq analysis. We used five somatic organs and three gonadal organ parts for each of the sexes, yielding a total of 576 samples (*Figure 1—source data 2*).

To identify sex-biased expression, based on normalized transcript fragment count data (transcripts per million [TPM]), we use the ratio of the medians of female (F) and male (M) expression (F/M) with a 1.25-fold ratio cutoff, combined with a Wilcoxon rank sum test and false discovery rate (FDR) correction <0.1. These criteria are more stringent than other studies that have focused on somatic sex-expression differences (see Methods for a discussion of these parameters). Note that the use of a Wilcoxon significance test acts as an additional filter to exclude genes with neutrally fluctuating variances between the sexes. The absolute fold differences between male and female expression are continuously distributed, at least for the organs with many sex-biased genes (see distribution histograms in *Figure 1—source data 3 and 4*). On the other hand, absolute fold differences should not necessarily be taken as an indicator of relative importance of the respective sex-biased genes, since absolute dosage differences can have different effects in different parts of the regulatory networks.

*Figure 1* provides a graphical overview of the numbers of sex-biased genes identified in the different organs of the mouse taxa; detailed numbers are provided in *Figure 1—source data 1*, the corresponding gene lists and expression values are provided in Supplementary Data D1. The numbers range between 13 and 5645 for the somatic organs and 3444 and 13,383 for the gonadal pairwise comparisons. As discussed above, the gonadal comparisons are inherently very different from the somatic comparisons, since they are based on comparing morphologically different structures (see Methods for the scheme of pairwise comparisons). Still, we include them as reference comparisons here, but the main focus of our study is on the somatic comparisons.

Variable numbers of sex-biased genes are shared between pairwise organ comparisons within the different taxa, but only very few genes are sex-biased in all organs in a given taxon (*Figure 1—source data 5*). These include female-biased genes such as the X-chromosomally encoded gene *Xist*, which codes for an lncRNA required for the dosage compensation of the X-chromosome in females (*Loda and Heard, 2019*), and *Eif2s3x*, which encodes the eukaryotic translation initiation factor 2 subunit 3 (*Xu et al., 2006*). Most other genes that share a sex-bias between all tissues in each taxon are also encoded on the X-chromosome, and most are known to escape the dosage compensation of the X-chromosome in tissue-specific patterns (*Berletch et al., 2015*; *Figure 1—source data 5*). Given that we find different sets of these genes in the different mouse taxa (*Figure 1—source data 5*), we can conclude that even these general function genes show some degree of evolutionary turnover in their regulation.

It has previously been reported that the X-chromosome harbors more female-biased than male-biased genes for the mouse somatic organs (*Reinius et al., 2012*). Our data confirm this general observation (numbers and percentages included in *Figure 1—source data 1*). When averaged across all five somatic tissues, the X-chromosome has a higher percentage of female-biased genes (4.3%

female-biased versus 2.3% male-biased), but given that the overall fraction of genes on the X-chromosome is 4.7%, this reflects a relative deficiency of male-biased genes on the X, rather than an excess of female-biased ones. The sex-biased genes of the gonadal organs show the opposite trend (3.7% female-biased versus 4.2% male-biased). However, there are substantial differences between the different organs and taxa in this respect, implying that there is no simple explanation for the evolution of sex-biased genes on the X-chromosome.

Female-biased genes outnumber male-biased genes in all comparisons, which could hint toward stronger intrasexual selection in females (see Discussion).

Among the somatic organs, brain has the lowest numbers of genes with sex-biased expression, and kidney has the highest. Most notably, the numbers of sex-biased genes are subject to fast turnover between the taxa for the somatic organs. For example, DOM has an unusually high number of sex-biased genes in the heart, compared to the other taxa. On the other hand, the kidney bias is much stronger in DOM and MUS than in SPR and SPI, and liver has the highest number of female-biased genes in SPI. For the gonadal organs, the overall numbers remain more similar between the taxa.

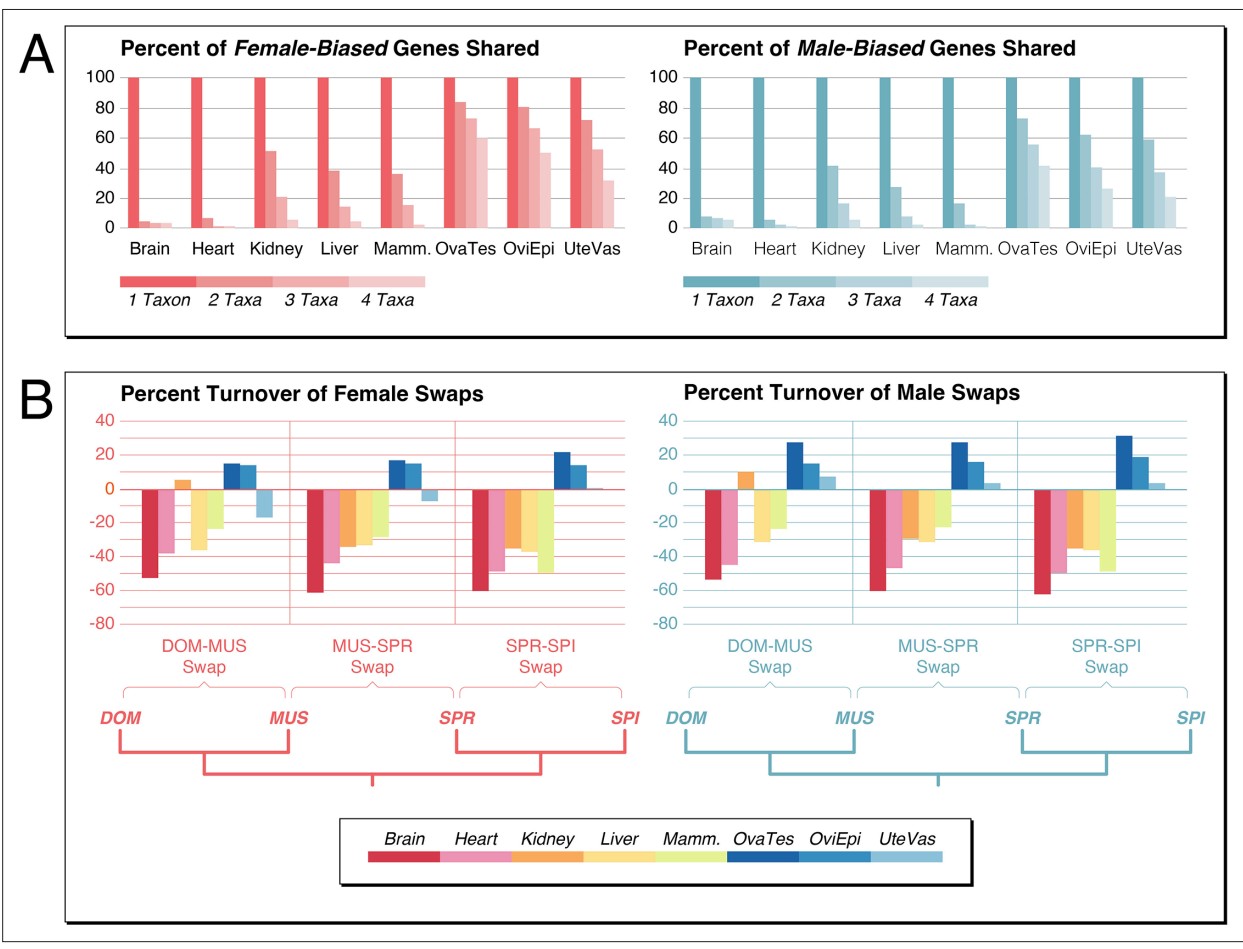

**Figure 2.** Turnover of sex-biased gene expression between the taxa. (**A**) Plots of percentages of genes shared as sex-biased across the four taxa for each organ comparison (including the Y-chromosomal genes for the male-biased gene sets). Numbers are normalized to 'one taxon' which represents the sum of all unique genes in at least one taxon (set to 100%), 'two taxa', 'three taxa', and 'four taxa' represent the percentages of the sums of shared genes for any pairwise comparison between the taxa for all sex-biased genes. Data for the figure are provided in *Figure 2—source data 1*. (**B**) Percentage turnover differences between sex-biased genes versus resampling averages from all genes as female gene swaps or male gene swaps in three groups of taxa comparisons. See text for further details. Note that standard deviations from the resampling were too small to show them in the graphic as error bars (all in the order of 0.015). Data for the figure are provided in *Figure 2—source data 2* and for all data and statistics in Supplementary Data D2.

The online version of this article includes the following source data for figure 2:

**Source data 1.** Numbers and percentages of sex-biased genes shared between taxa for each organ.

**Source data 2.** Comparative analysis of gene expression level turnover between the taxa.

## Fast evolutionary turnover of sex-biased gene expression

The evolutionary turnover of genes that become subject to sex-biased expression in any given taxon is very high, even between the closely related taxa that are studied here. *Figure 2A* shows which fractions of genes with sex-biased expression are shared between the taxa for each organ in any pairwise comparison. The turnover is particularly high for the somatic tissues. In brain and heart, fewer than 10% of the genes occur in more than one taxon; for the other organs, fewer than half occur in more than one taxon. Note that two of the taxa (DOM and MUS) have a subspecies-level status (*M. m. domesticus* and *M. m. musculus*), yet the majority of somatic sex-biased genes changed their sex-biased expression between them. When compared between all four taxa, only between 0.3% and 5.3% of genes have a conserved sex-biased expression in all four of them (*Figure 2A*).

The fractions of shared genes are higher for the gonadal tissues (*Figure 2A*), but even these tissues show a substantial turnover of sex-biased genes. Interestingly, many of the sex-biased genes reverse their state in at least one taxon from female-bias to male-bias or vice versa. This includes 596 genes for somatic organs and 3895 for the gonadal organs (*Figure 1—source data 1*). Hence, although the overall numbers of sex-biased genes do not change much between the taxa for the gonadal organs, the actual composition of the gene sets evolves substantially.

To compare the rates of this turnover of sex-biased gene regulation with non-sex-biased regulatory turnover, we use the same statistical approach for the latter group of genes, but instead of comparing males with females, we use a comparison between the same sex datasets between the taxa. For example, for a turnover comparison between DOM and MUS, we swapped the DOM males with the MUS females in the same statistical algorithm that we use for the sex-bias analysis. We did this also for the SPR-SPI and the MUS-SPR pairs, as well as the reverse sex groups, i.e., a dataset with comparison among females only ('Female Swaps') and a dataset with comparisons among males only ('Male Swaps'). From the resulting genes lists, we resampled 1000 times 1000 genes each and calculated averages and standard deviations. The corresponding data and statistics tables are provided as Supplementary Data D2. The percentages of sex-biased genes that show a regulatory change were then subtracted from the average percentages of the equivalent all-genes comparisons in the same taxa groupings. Hence, when sex-biased gene expression turnover is higher than the all-genes expression turnover, one should find negative values. The results are shown in *Figure 2B*. Except for the kidney in the DOM-MUS comparison, all somatic organs in all comparisons show negative values, while the gonadal organs in all comparisons mostly show positive values.

This analysis shows that somatic sex-biased gene expression changes occur much more frequently than for the whole gene set. Only the kidney in the DOM-MUS comparison is very different in this comparison, possibly since they have been subject to other rapid evolutionary processes that are currently unknown. The positive value for kidney in the DOM-MUS comparison is mostly due to a relatively high number of sex-biased genes that are conserved between these two taxa, for as yet unknown reasons. This pattern is different for the two other taxa comparisons, where sex-biased expression evolves also faster.

The gonadal turnover of sex-biased gene expression, on the other hand, is not faster than the gene expression levels of the non-sex-biased genes in most comparisons. This implies also that the sex-biased status in the gonads has a more conserved function, most likely since it is linked to different cell types in these tissues.

## Variance patterns

The fast turnover of genes with sex-biased expression implies that they are continuously subject to intra- or inter-sexual selection processes. This should also be reflected in the variance patterns within the populations of each taxon, since at least a subset of them would be subject to ongoing selection processes that would create a higher allelic diversity until new equilibria are reached. We have therefore analyzed the variance patterns both at the gene expression level and at the protein divergence level.

As a measure of variance for the gene expression, we used the interquartile range (IQR) (difference between the 75th and 25th percentiles of the data) divided by the median of the data range (representing the nonparametric version of the coefficient of variation). This measure buffers against outlier values that can frequently be observed in the data (*Xie et al., 2025*). For these comparisons, we also used a set of non-sex-biased genes, defined as genes having an F/M ratio of <1.05-fold.

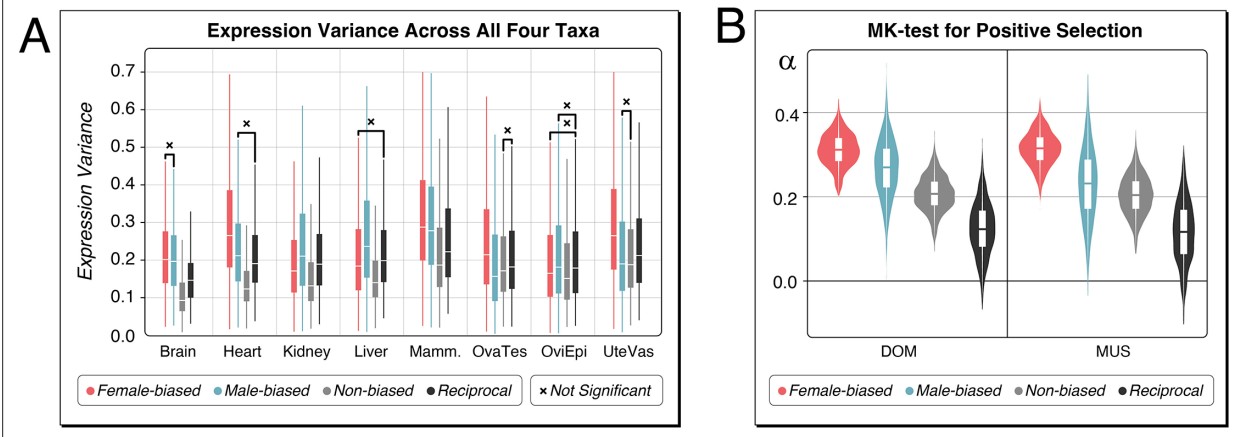

**Figure 3.** Variances and positive selection on sex-biased genes. (**A**) Variances of expression in sex-biased and non-biased genes for each organ. The ranges of relative variances (interquartile range [IQR]/median ratios for transcripts per million [TPM] counts) for the four taxa are displayed as box plots; note that these constitute all values from the four taxa. The 'reciprocal' values are for the orthologous genes that are sex-biased in one taxon, but not in the other for *M. m. domesticus* (DOM)-*M. m. musculus* (MUS) comparisons and *M. spretus* (SPR)-*M. spicilegus* (SPI) comparisons. The data for this sub-figure are provided in *Figure 3—source data 1*. Most pairwise comparisons are significant (p<<0.01), the ones which are not significant are marked with 'x' (all p-values are included in *Figure 3—source data 1*). (**B**) Results of the McDonald-Kreitman (MK) test for positive selection at coding positions for the sex-biased genes in DOM and MUS. The 'reciprocal' values are for the orthologous genes that are sex-biased in one taxon, but not in the other. Note that this corresponds to different gene sets in DOM and MUS, since they have different sets of sex-biased genes each. The alpha values represent the fraction of amino acid substitutions that are predicted to be driven by positive selection. The violin plots are derived from the range of alpha values obtained in 1000 bootstrap replications. The boxes show the averages and quartiles of the data distribution. The averages between all distributions are significantly different from each other (p<<0.01). All data, including the gene numbers in the analysis and statistical values, are provided in *Figure 3—source data 2*.

The online version of this article includes the following source data for figure 3:

**Source data 1.** Data for expression variance across all four taxa.

**Source data 2.** Results of the McDonald-Kreitman (MK) test with 1000 bootstrap replications.

We find that sex-biased genes show in most cases higher variances than non-biased genes, but to different extents in the two sexes and in the different organs (*Figure 3A*). The pattern is much more distinct for somatic genes than for gonadal genes, in line with the findings of the overall divergence patterns described above.

To assess whether sex-biased genes are recruited from genes with higher general expression variances, we created reciprocal gene sets with the orthologous sex-biased and non-sex-biased genes in pairwise taxa comparisons (DOM-MUS and SPR-SPI). These show variance levels more comparable to those found among the sex-biased genes (*Figure 3A*), implying that their generally higher variation might help them to become more easily sex-biased, as it has also been suggested for plant sex-biased genes (*Scharmann et al., 2021*).

Previous studies on protein sequence divergence patterns of sex-biased genes have suggested that the sequences of male-biased genes are particularly fast evolving (*Grath and Parsch, 2016*; *Harrison et al., 2015*), while the opposite pattern was found for the avian brain (*Mank et al., 2007*). Sex-biased genes in sex-dimorphic plant leaves, on the other hand, did not show increased divergence compared to non-sex-biased genes (*Scharmann et al., 2021*). However, all the previous tests on this question were somewhat limited by not taking possibly confounding population demographic factors into account. We have therefore included the population-level information in our data to address this question.

A frequently used test for testing for protein-level divergence in population data is the McDonald-Kreitman (MK) test. It compares the level of polymorphism versus fixation of coding and noncoding sites to infer the frequency of positive selection events (*McDonald and Kreitman, 1991*). It calculates an alpha value that is a measure of the proportion of amino acid substitutions that were driven by positive selection rather than genetic drift. However, its results can be influenced by confounding factors, such as the level of segregation of slightly deleterious variants, as well as population demography. We have therefore used here the asymptotic MK test that takes care of these problems (*Haller*

*and Messer, 2017*; *Messer and Petrov, 2013*), but requires the testing across a large number of loci to become reasonably robust. We have therefore combined the whole set of sex-biased genes of the somatic organs for a given taxon in the test. We restrict the comparison to the DOM and MUS samples, since these are closest to the reference sequence to ensure appropriate alignments of the coding regions to correctly assign coding and noncoding sites. The polymorphism data were taken from previous whole-genome sequencing studies of these populations (*Harr et al., 2016*). Classification of fixed versus polymorphic variants was done by comparing to SPR as an outgroup. Statistical estimates are based on 1000 bootstrap replications.

The results are shown in *Figure 3B*. Both female-biased and male-biased genes show higher alpha values compared to non-biased genes. Interestingly, female-biased rates are higher than male-biased rates, corroborating the observation in brain-expressed genes in birds (*Mank et al., 2007*). Similar to the expression variance comparisons, we also generated reciprocal gene sets, i.e., sex-biased in either DOM or MUS, but not sex-biased in the respective other taxon. The alpha values for the non-sex-biased reciprocal orthologs are significantly lower (*Figure 3B*) than those for the sex-biased genes. This comparison suggests that the evolution to a sex-biased expression state is often accompanied by an enhanced adaptive substitution rate at coding positions of the respective genes.

To assess whether we can also trace lineage-specific effects with this test, we asked whether the genes that are exclusively sex-biased in either DOM or MUS also show higher alpha values. However, for this test, we needed to combine female- and male-biased genes to have a sufficiently large number of genes to render the test meaningful. We find that the genes that are sex-biased only in DOM have an alpha value of 0.22, while the non-sex-biased orthologs of these genes in MUS (the 'reciprocal' gene set) have an alpha value of 0.12. The corresponding numbers for MUS are 0.14 and 0.11 (see *Figure 3—source data 2*). This contrasts with alpha values of 0.35 and 0.37 for genes that are sex-biased in both taxa. Hence, we can indeed trace signs of a lineage-specific elevation of adaptive substitutions in genes that become sex-biased, although the effect is stronger when they were sex-biased for a longer evolutionary time.

## Module analysis of sex-biased genes

We were interested to assess whether sex-biased genes represent a subset of all genes expressed in a given organ or whether they are derived from particular co-regulated modules. The summary results are provided in *Figure 4*. We used weighted gene co-expression network analysis (WGCNA) (*Langfelder and Horvath, 2008*) to determine gene expression modules for each organ, based on a set of transcriptomes from 48 DOM females that were prepared in parallel to the remainder of the samples, because WGCNA requires such large sample sets to generate reasonably robust results. The analysis of the scale independence and mean connectivity plots for the different organs in this dataset shows that saturation is reached at levels of soft power β between 4 and 8 (*Figure 4—source data 2*; see also Methods). Since a similarly large set is currently not available for males, we restrict the further analysis to the female-biased genes, which actually constitute the larger fraction of sex-biased genes between the sexes.

The individual module assignments for all genes in each of the five somatic organs are listed in *Figure 4—source data 3*. We found between 21 and 64 modules in the different organs. The numbers of genes in each module for each taxon and tissue are provided in *Figure 4—source data 4* (note that module number assignments are not comparable between the organs - they are always only determined for a given organ). The comparisons show that the sex-biased genes are not simply a proportional reflection of all modules, but are enriched for some subsets of modules. To directly visualize this, we subtracted the normalized fraction of gene numbers in each module from the normalized fraction of sex-biased genes for each taxon. A subset of the data is plotted in *Figure 4*. This subset is restricted to four somatic organs, since brain has too few sex-biased genes overall. Further, we did not include higher-numbered modules with few genes only, unless the differences to the fractions of all genes are more than 3% in the sex-biased genes.

The results suggest that the module classes for the sex-biased genes tend to be retained between the taxa, despite the turnover of genes. This is further detailed below for each organ.

For heart, there is an excess of genes in module 1 for DOM, MUS, and SPR. Module 1 includes 3010 genes associated with multiple physiological processes in a GO analysis. DOM includes 1247 of these among the total of 2405 sex-biased genes (=52%). MUS includes 171 out of 337 (=51 %) and

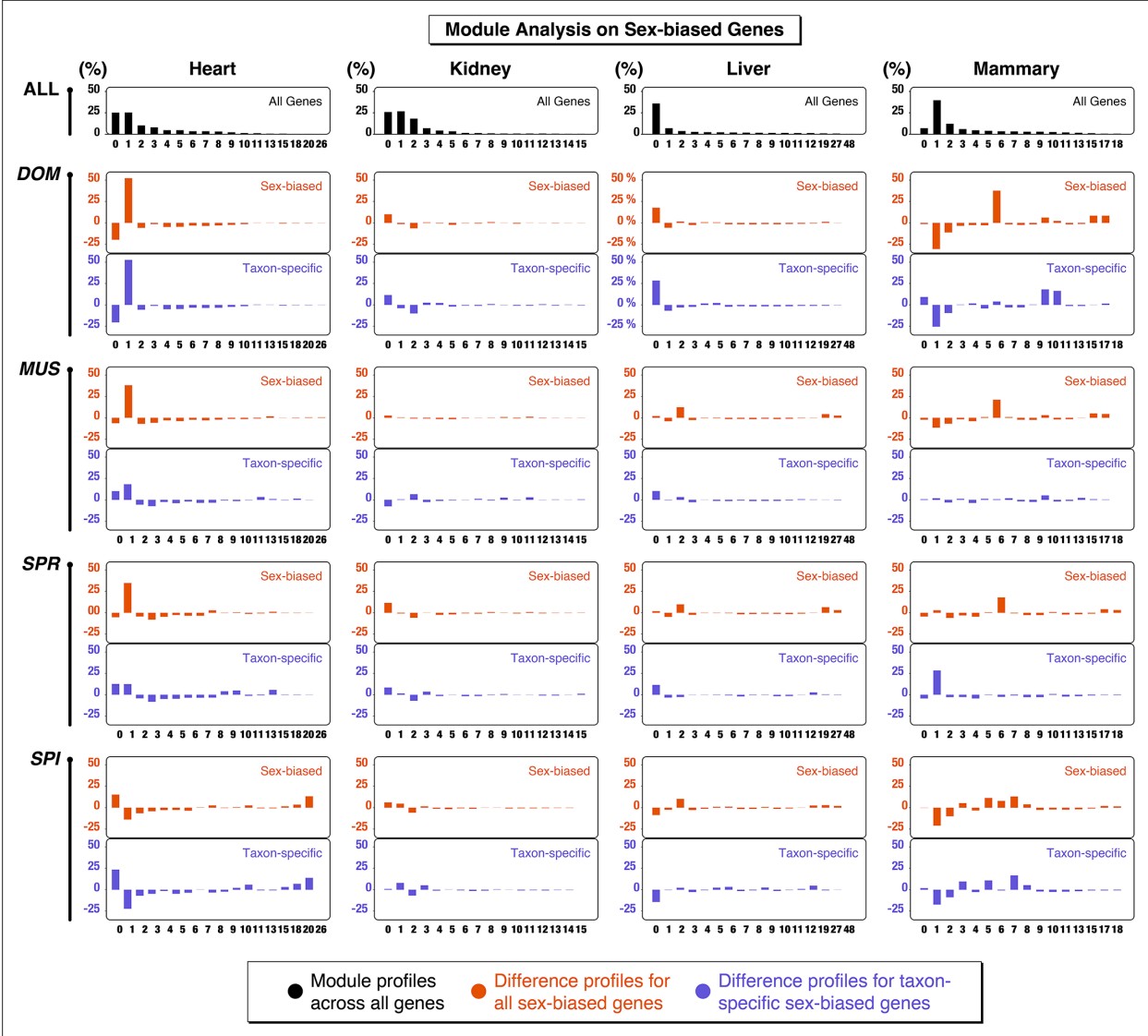

**Figure 4.** Module analysis on sex-biased genes. The plots show for four somatic organs the fractions of total genes (row 'all' at the top - black color) that are assigned to modules in a weighted gene co-expression network analysis (WGCNA). The rows below represent the fraction differences of sex-biased genes for each taxon (DOM, MUS, SPR, SPI) compared to all genes. Positive values show an excess and negative values a deficiency compared to the fraction of all genes. Plotted in red are the differences for all sex-biased genes in the organ, in blue for the sex-biased genes that occur only in the respective taxon, i.e., can be considered as having a newly evolved sex-bias expression in this taxon. A maximum of 15–16 modules plus the 0 bin are plotted for each organ. Higher-numbered modules are plotted only when they include a difference of larger than 3% in at least one taxon. The full data are provided in *Figure 4—source data 1*. Note that the module numbers can only be compared within an organ, not between the organs. Bin number '0' is the sum of all genes that cannot be assigned to one of the other modules. The brain is not included in this figure, since it has too few sex-biased genes to make the comparison meaningful. DOM, *M. m. domesticus*; MUS, *M. m. musculus*; SPR, *M. spretus*; SPI, *M. spicilegus*.

The online version of this article includes the following source data for figure 4:

**Source data 1.** Data for module plots with normalized gene counts - reduced set, see text.

**Source data 2.** Scale independence and mean connectivity plots for the different organs for determining the soft threshold parameter β for the weighted gene co-expression network analysis (WGCNA) module assignments.

**Source data 3.** Weighted gene co-expression network analysis (WGCNA) module lists for all genes in all somatic organs derived from analyses between 48 females.

**Source data 4.** Gene counts for each module class in all somatic organs.

SPR 30 out of 76 (=39%). In MUS, 54 out of the 171 sex-biased genes in module 1 are not sex-biased in DOM, and in SPR, there are 6 sex-biased genes in module 1 that are neither present in DOM nor in MUS. Hence, there is a substantial turnover of sex-biased genes within module 1 between the taxa. In SPI, on the other hand, there is mostly a relative loss of sex-biased genes in module 1 (it retains only 6 such genes), but a relative major gain in a new module. This is module 26 with a total of 20 genes, 7 of which occur in SPI as sex-biased genes (4 are transcription factors involved in circadian clock entrainment). The profiles for the taxon-specific sex-biased genes (blue) are very similar to all sex-biased genes (red) in each comparison, implying that both are parts of the same modules in each taxon.

For kidney, the patterns are more heterogeneous, with little commonality between the taxa. Also, the patterns between all sex-biased genes and that taxon-specific sex-biased genes differ for some of the modules.

For liver, it is mostly module 2 that is enriched for sex-biased genes. The module has a total of 389 genes, which are enriched for innate immunity regulation GO terms. Here, MUS and SPI have 47% and 52% the largest fractions of sex-biased genes in this module. The patterns of the recruitment of taxon-specific sex-biased genes are somewhat heterogeneous, and module 2 is not specifically notable.

For mammary, it is module 6 that is most strongly enriched for sex-biased genes. The module has 540 genes, which relate mostly to epithelial cell morphology and tubule formation. MUS and DOM have the largest fractions of sex-biased genes in this class, with 83% and 73%, respectively. Again, the patterns for the taxon-specific sex-biased genes are somewhat heterogeneous; module 6 is not specifically notable. But the changed overall pattern in SPI is evidently driven by genes newly recruited to become sex-biased in a subset of modules in this taxon.

We conclude from this analysis that sex-biased genes tend to be expressed in a subset of modules and that evolutionary turnover may sometimes occur by recruiting previously not sex-biased genes from the same module to become sex-biased. However, this pattern is clearly heterogeneous between organs and taxa. It applies to some comparisons, but not to all.

## A sex-biased gene-expression index (SBI)

Given that the sex-biased expression of genes is thought to be the basis for the generation of the sex-specific phenotypes, we were interested in assessing the individual variation in the cumulative expression of sex-biased genes for each organ. To do this with a single factor for every individual, we have developed an SBI based on the normalized expression values of the genes. This is calculated as:

$$\text{SBI} = (\text{MEDIAN of all female-biased TPM}) - (\text{MEDIAN of all male-biased TPM})$$

For the male-biased genes, we exclude the ones encoded on the Y-chromosome, since they have no equivalent in females. The MEDIANs are used to reduce the influence of outlier values (*Xie and Tautz, 2025*).

When the sexes are well separated, one would expect two different distributions for each sex. The centers of the distribution can move along the X-axis, depending on the relative expression levels on males and females. While this provides a measure of the overall expression levels of sex-biased genes in the organs, this may in itself be of less relevance. For example, in plants, it was shown that the sex-biased morphological variation of the leaves does not correlate with the numbers and expression of sex-biased genes in these leaves (*Scharmann et al., 2021*). But independent of considering the numbers and absolute expression levels of the genes, one can look at the variance of expression between the individuals for each organ and how this variance evolves between the taxa. Such a comparison has not been done before in any dataset, and the SBI is uniquely suitable to visualize such comparisons (see also Discussion).

To focus on the variance aspect of the SBI, we normalize the values to center the distributions around 0. In this display, the females should show a range of positive values representing their individual variation, and the males should show an analogous range of negative values. But the variation ranges could also overlap, which would indicate that the phenotypes generated by the set of sex-biased genes are not completely sex-specific, similar to most somatic phenotypes, such as height measures or brain structures are not completely sex-specific (*Joel, 2021*; *Maney, 2016*).

The normalized distributions of individual sex-bias indices are plotted as density functions across all mouse organs and taxa in *Figure 5*. We find indeed overlapping patterns in most comparisons of the somatic organs. However, the organs differ with respect to their overlap patterns, and these

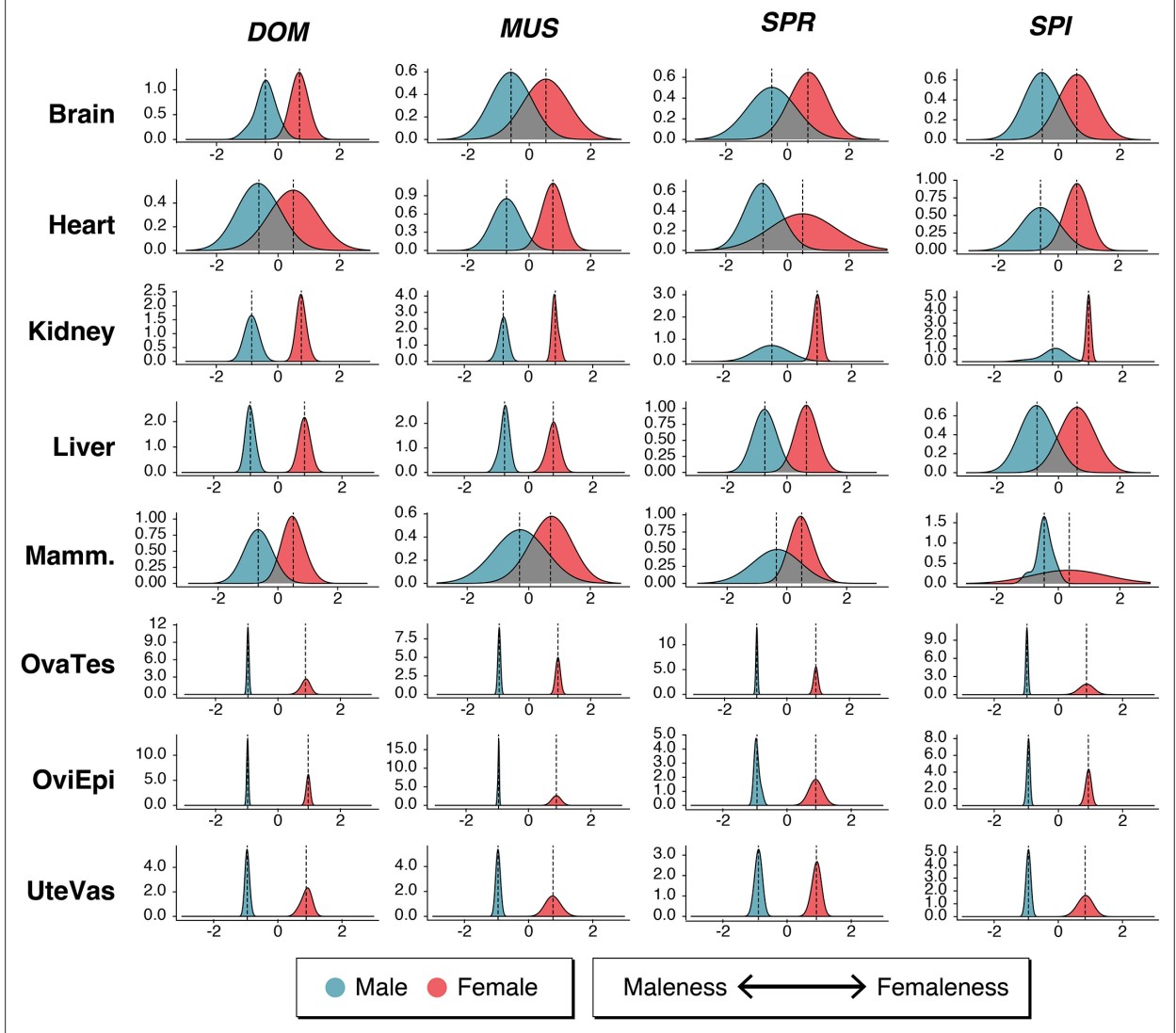

**Figure 5.** Density plots of individual variation values of the sex-biased gene expression index (SBI) for mouse organs. Plots for all organs are grouped according to organ for each taxon. Males are represented by blue shading, females by red shading. The taxon designations are on the top, the organ designations to the left. The Y-axis represents the density scale of the smoothed distribution; the X-axis represents the relative maleness <--> femaleness scores centered around zero. All individual SBI values are included in *Figure 5—source data 1*.

The online version of this article includes the following source data for figure 5:

**Source data 1.** Sex-biased gene expression index (SBI) values for mouse tissues.

**Source data 2.** Spearman's correlation test values between organ sex-biased gene expression indices (SBIs).

differences evolve fast between the taxa. This is particularly evident for kidney and liver. Kidney shows nonoverlapping distributions in DOM and MUS, while in SPR and SPI, the male-biased genes show a very broad distribution that overlaps with the distributions of female-biased genes. Strong sexual dimorphism in mouse kidneys and livers was previously described and is linked to hormonal signaling pathways, especially in males (*Clodfelter et al., 2006*; *Laulhé et al., 2021*). In liver, DOM and MUS are also well separated, while SPR and SPI are overlapping. For mammary, the overlaps are also substantial in all four taxa, although one might have predicted that they could be more distinct. Such a distinctness is evident for all comparisons between gonadal organs, but these also generate very different morphological structures, i.e., an overlap is not expected.

Given that the same sets of organs were retrieved from each individual, it is possible to ask whether SBIs might correlate between organs of the same individual. For example, an individual with a strong femaleness score in one organ might also have a strong femaleness score in another

organ. However, this is not the case. There is no significant correlation in any pairwise comparison between organs for a given sex (Spearman's rank correlation test, all p-values with multiple testing correction >0.05) (*Figure 5—source data 2*). This implies that the femaleness and maleness status of an individual is not homogeneous throughout the body - each organ can be somewhat different in this respect.

## Mosaic patterns of sex-biased gene expression

The comparative analysis of sex-biased brain structures based on MRI datasets of more than 1400 human brains has suggested that brains are usually composed of a mosaic of sex-biased structures (*Joel et al., 2015*; *Joel et al., 2020*). The above correlation analysis of SBI values between mouse organs has also suggested that such a mosaic pattern could apply (see above). To visualize this more directly, we use the heatmap approach developed by *Joel et al., 2015*, to plot the SBI values for each individual and organ via a normalized color scale (*Figure 6*). These plots show that the SBIs differentiate the individuals such that each has a more or less unique combination. While the patterns of males and females are distinct, they also show overlaps for individual values in the middle ranges.

## Sex-biased gene expression in humans

The SBI can be applied to any comparative transcriptome data of population samples from both sexes. The GTEx consortium has generated such data for humans (*Aguet et al., 2020*) and these have previously been analyzed with respect to sex-specific expression patterns (*Gershoni and Pietrokovski, 2017*; *Khodursky et al., 2022*; *Oliva et al., 2020*). Unfortunately, these data are much more heterogeneous than the mouse data, with variable age distributions, data quality, and death reasons. Various statistical procedures are therefore usually employed to control for confounding variables (*Aguet et al., 2020*; *Khodursky et al., 2022*; *Oliva et al., 2020*; *Wolf et al., 2023*). The largest problem is the overdispersion of the data, including the frequent occurrence of outlier values. The procedure that we have used for the mouse data addresses these problems stringently (see Methods), and we therefore apply it to the human data as well. For comparison, we also use the lists of the sex-biased genes generated by the GTEx consortium that were generated with a more permissive procedure across all samples (*Oliva et al., 2020*). We focus here on individuals younger than 49 who did not die after a long disease phase. These individuals are relatively rare in the GTEx data, given the strong bias for older individuals in the dataset. But we could retrieve nine females and males each for most organs and organ subsets for the analyses shown in *Figure 7* (the full set of data includes 27 female-male comparison sets; *Figure 7—source data 2*; Supplementary Data D3), making the patterns directly comparable to the mouse patterns. Our overall results are qualitatively very similar to the previously published results on these data, but given our more stringent filtering due to including an FDR step and an explicit cutoff, the overall numbers are lower, especially in comparison to *Oliva et al., 2020*; *Figure 7—source data 3*.

Compared to the mouse, we find generally fewer sex-biased genes in most human tissues. Among 27 tissues included, only 10 have at least five sex-biased genes in either sex (*Figure 7A*). The tissues with the largest numbers of sex-biased genes are 'Adipose Subcutaneous' and 'Breast Mammary Tissue' (*Figure 7B* - note that the Y-axis scale of this figure is 15-fold expanded compared to *Figure 7A*). Further, the comparison between ovary and testis (OvaTes) between males and females shows large numbers of sex-biased genes, but the tissues have evidently also very different cell compositions for these organs.

The calculation of the SBI is limited when there are only very few sex-biased genes in one or both tissues. We are therefore plotting the SBI only for tissues with more than five sex-biased genes for either sex (*Figure 7C*). All of the SBI distributions are more or less overlapping between males and females, including breast tissue, similar to that in mouse. Only the comparison between ovary and testis (OvaTes) is very distinct, as to be expected.

When using the lists of sex-biased genes from *Oliva et al., 2020*, to calculate SBIs for the same individuals and same organs, we find qualitatively similar results but an even higher overlap between the sexes (*Figure 7D*). This is due to the inclusion of many additional genes that fall below our threshold of 1.25-fold change (compare numbers in *Figure 7—source data 3*).

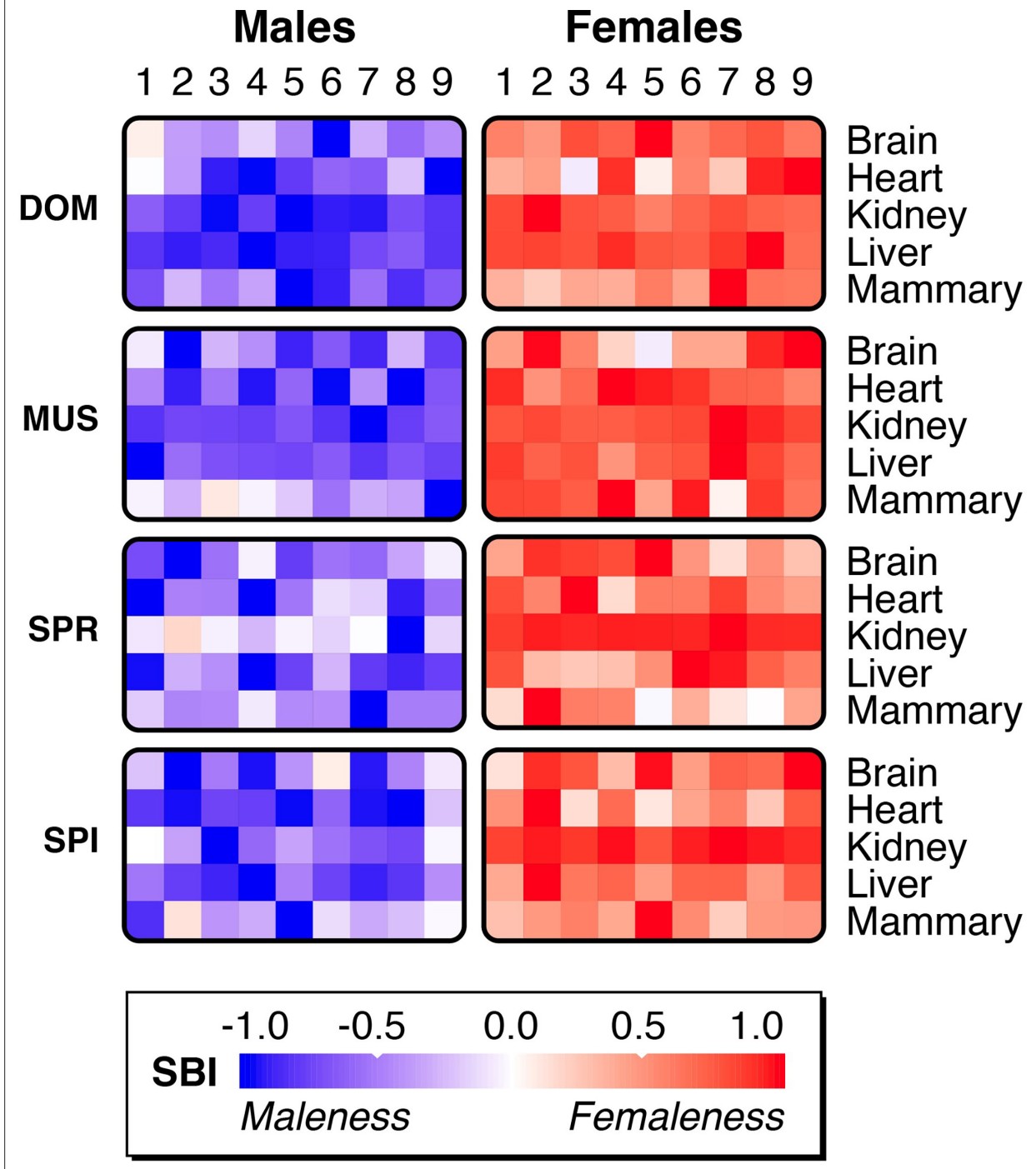

**Figure 6.** Heatmap plots of sex-biased gene expression index (SBI) values for mice. Each individual (numbered 1–9 for each sex) is represented by the normalized SBI values for the somatic organs organized in rows. The color scale represents the normalized SBI value in the range from 1 (maximal femaleness, dark red) to –1 (maximal maleness, dark blue). The mouse data are provided for the individuals from all four taxa. All individual SBI values are included in *Figure 5—source data 1*.

## Sex-biased expression in human single-cell data

Gene expression in organs is measured from complex aggregations of diverse cell types, making it difficult to distinguish between sex differences in expression that are due to regulatory rewiring within similar cell types and those that are simply a consequence of developmental differences in cell-type abundance (*Darolti and Mank, 2023*). It has actually been shown that only a subset of cells in a

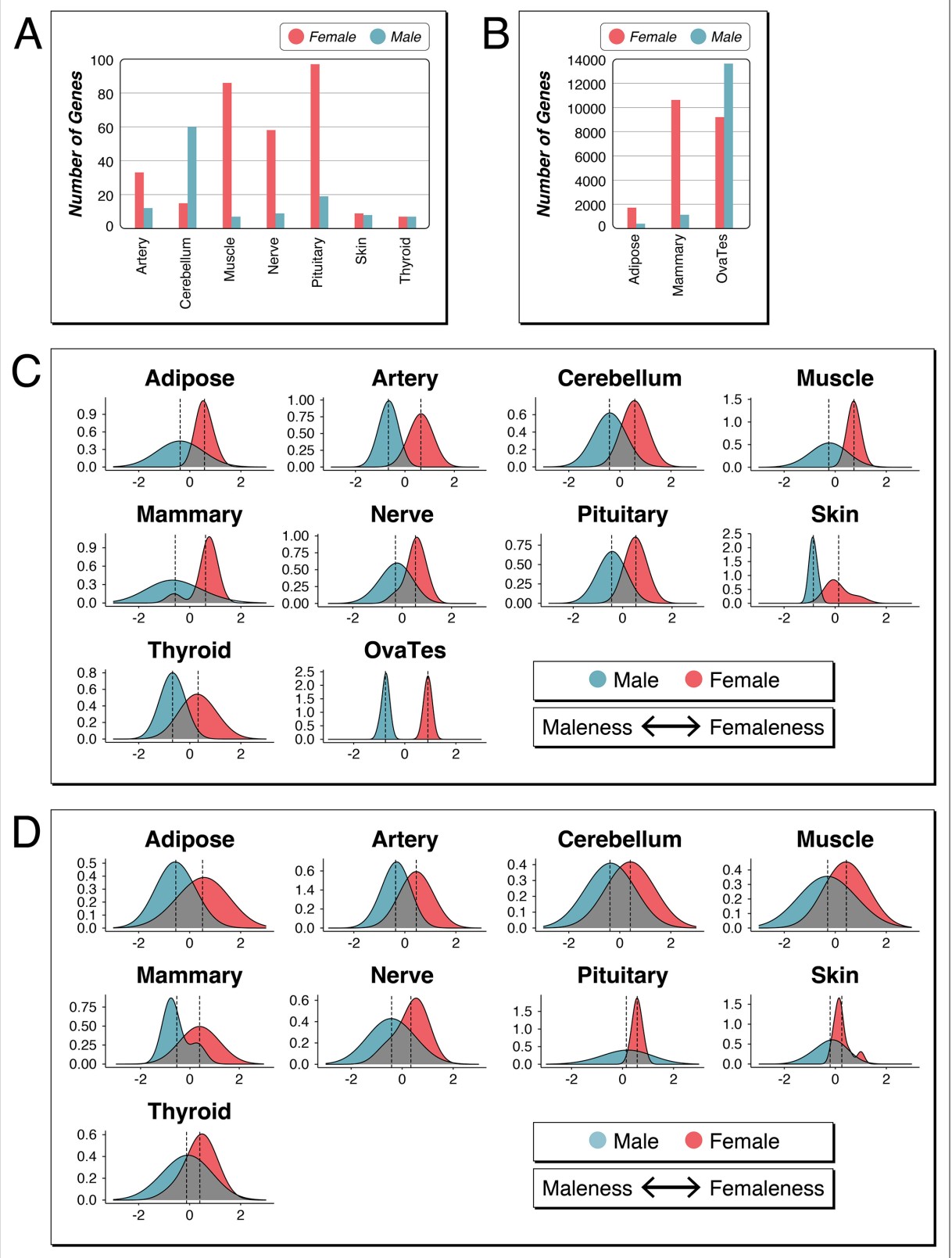

**Figure 7.** Sex-biased genes and sex-biased gene expression indices (SBIs) for data from human tissues. (**A**) and (**B**) Bar plots representing the numbers of sex-biased genes in tissues that show at least five such genes per sex (excluding the Y-chromosomally encoded genes). Note the different Y-axis scale for the numbers of genes in (**B**). (**C**) SBI plots for the nine individuals of each sex based on the set of organs and genes shown in (**A**) and (**B**). The Y-axis represents the density scale; the X-axis represents the relative maleness <-> femaleness scores centered around zero. (**D**) SBI plots for the

*Figure 7 continued on next page*

Figure 7 continued

same individuals and organs as in (**C**), but based on the sex-biased gene lists from *Oliva et al., 2020*. Note that these authors have not included a comparison for OvaTes. All SBI values are listed in *Figure 7—source data 1*.

The online version of this article includes the following source data for figure 7:

**Source data 1.** Sex-biased gene expression index (SBI) values for human tissues.

**Source data 2.** List of samples used for the analysis of sex-biased expression in humans.

**Source data 3.** Numbers of sex-biased genes found in the human organ samples.

given tissue may express the sex-biased genes (*Rodríguez-Montes et al., 2023*). In complex organs with many cell types, such as the brain, it is therefore possible that the sex-bias signal is blurred by different sets of genes expressed in different cell types. Unfortunately, suitable datasets from single-cell studies that include sufficient numbers of individuals where one could approach this question are still rare. We have analyzed here data from a study with patients that have developed Alzheimer's disease. This study also included non-disease control individuals, and we have used the datasets from these individuals, yielding comparisons between six individuals of each sex for single cells from dorsolateral prefrontal cortex (DLPFC) samples and seven individuals of each sex for single cells from middle temporal gyrus (MTG) samples (*Gabitto et al., 2024*). There are 15 different cell types with sufficient coverage for an analysis in the DLPFC data and 13 in MTG data (*Table 1*; *Table 1—source data 1*). A subset of 12 cell types overlaps and can therefore be compared between the tissues with the results shown in *Table 1* and detailed in *Table 1—source data 1*. Sex-biased expression is still at a low level in these data, similar to what was found for brain tissues in general and too low to generate meaningful SBI distribution plots. While this may be partly due to the age of the individuals and/or insufficient coverage of low-level expressed genes in single-cell datasets, it still implies that there is not any particular cell type with very strong male-female differentiation.

## Conserved genes with sex-biased expression

While most studies on sex-biased genes have reported that only a subset of them is conserved across larger evolutionary distances, they still report sometimes substantial numbers of genes with

**Table 1.** Numbers of genes with sex-biased expression in human single-cell data.

| Cell type | Female-biased | | | Male-biased* | | |
|---|---|---|---|---|---|---|
| | MTG | DLPFC | Overlap | MTG | DLPFC | Overlap |
| Astrocyte | 4 | 7 | 4 | 2+4 | 0+4 | 4 |
| L2_3_IT | 4 | 1 | 1 | 0+4 | 1+4 | 4 |
| L4_IT | 1 | 1 | 1 | 0+5 | 0+5 | 4 |
| L5_IT | 1 | 1 | 1 | 0+5 | 0+4 | 4 |
| L6_IT | 1 | 2 | 1 | 0+4 | 2+4 | 4 |
| Lamp5_Lhx6 | 2 | 1 | 1 | 0+5 | 0+5 | 5 |
| Microglia-PVM | 5 | 10 | 4 | 1+6 | 7+6 | 6 |
| Oligodendrocyte | 4 | 6 | 3 | 1+4 | 0+4 | 4 |
| OPC | 7 | 3 | 2 | 0+5 | 0+5 | 5 |
| Pvalb | 2 | 1 | 1 | 0+5 | 0+5 | 5 |
| Sst | 1 | 7 | 1 | 0+5 | 0+5 | 5 |
| Vip | 1 | 1 | 1 | 0+5 | 0+5 | 5 |

*Non-Y-chromosomal+Y-chromosomal.

The online version of this article includes the following source data for table 1:

**Source data 1.** Comparison of genes shared in the single-cell data between the two brain tissues, dorsolateral prefrontal cortex (DLPFC) and middle temporal gyrus (MTG).

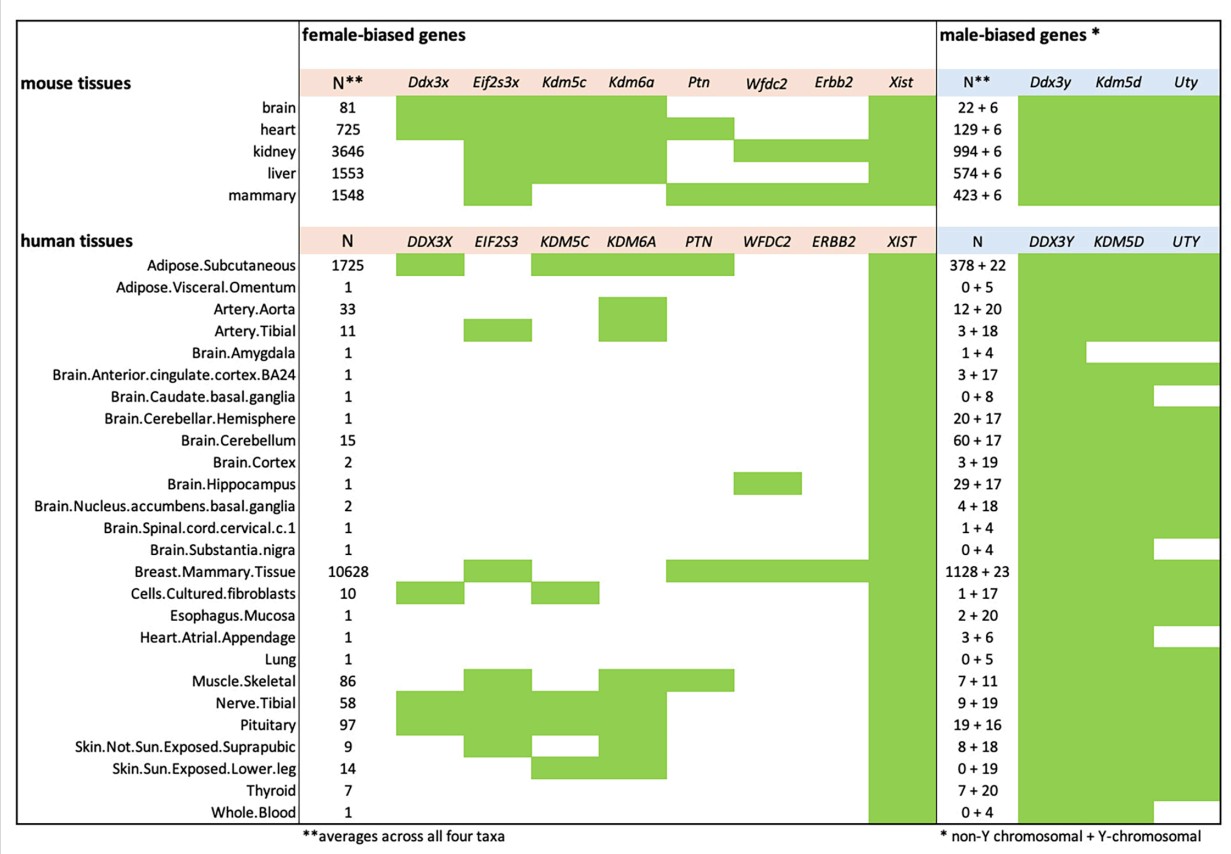

**Figure 8.** Genes with conserved sex-biased gene expression between mice and humans. The different organs analyzed are listed to the left. The analysis of conserved mouse genes is included in *Figure 8—source data 1*. The green boxes designate sex-biased expression in the respective tissues.

The online version of this article includes the following source data for figure 8:

**Source data 1.** Mouse sets of sex-biased genes that occur across all four taxa for each organ.

conserved sex-biased expression (*Harrison et al., 2015*; *Khodursky et al., 2022*; *Naqvi et al., 2019*). In contrast, the study by *Rodríguez-Montes et al., 2023*, found much fewer conserved sex-biased genes between mammalian taxa but a conservation of cell types with sex-biased expression in each tissue. Given the high turnover of genes that we observe already within mouse lineages that are only at most 2 million years apart, it seems well possible that the seeming conservation of genes with sex-biased expression between lineages with larger evolutionary distances is due to convergence, i.e., independent recruitment of genes into sex-biased expression in different lineages. This has also been suggested in a phylogenetic study on sex-biased genes in plants (*Scharmann et al., 2021*).

We have therefore sought to generate a set of genes with conserved sex-biased expression using stringent criteria. First, we retrieved all genes that show a consistent sex-biased expression in all four mouse taxa. We then focused on those genes that occur in at least two of the analyzed tissues. This filters for genes that can be expected to have a more general function in regulating sex-biased expression. Based on this set, we asked which orthologs of the genes also show sex-biased expression in any of the human tissues. Interestingly, only very few genes are left when applying these filters (*Figure 8*). There are eight orthologous genes that are female-biased in two or more tissues and three that are male-biased. Note, however, that the latter are all genes that are encoded on the Y-chromosome. The only two genes with consistent sex-biased expression across all organs and species are the female-biased gene *Xist* and the male-biased gene *Ddx3y*.

Four of the conserved genes (*Kdm5c, Kdm5d, Kdm6a,* and *Uty*) are involved in histone methylation and demethylation, thereby triggering gene regulatory cascades that are known to be involved in setting up the sex-specific expression networks (*Deegan et al., 2019*; *Samanta et al., 2022*). Hence, the balancing of the interplay between these epigenetic regulators may well contribute to the fast

evolution of sex-biased gene expression in their target genes. However, while all four are broadly expressed in adult tissues, they are themselves subject to differential sex-biased expression in a subset of the tissues (*Figure 8*), implying that they also depend on sex-bias controlling expression networks.

## Discussion

The present study is the first micro-evolutionary study on the evolution of sex-biased gene expression in outbred animals that includes sufficient data to allow within-population comparisons in parallel to phylogenetic comparisons between closely related taxa. Further, its focus is on somatic sex-biased expression of genes, which is of special relevance for understanding the variances of sex-related phenotypes of adult individuals in populations. The data show that somatic sex-biased gene expression evolves very fast, even between the closely related taxa studied here. Moreover, many genes reverse their role by switching between their sex-bias to the other sex. Similar results were found for the evolution of sex-biased genes in the *Drosophila* brain (*Khodursky et al., 2020*) and in sex-biased genes expressed in sexually dimorphic leaves of plants (*Scharmann et al., 2021*). Variances of sex-biased gene expression between individuals are often overlapping between the sexes, which can be shown by plotting SBI values.

### Neutral or adaptive?

Patterns of fast evolution of molecular characters such as gene expression or substitution rates always raise the question of which fraction could be due to neutral divergence versus adaptive processes, i.e., positive selection for new variants. This question has previously been studied in the same system of mouse populations and taxa that are used here. These studies showed that the overall patterns of gene expression changes between these populations are mostly compatible with neutral divergence models, based on tests of intra-group variability with between-group divergence (*Bryk et al., 2013*; *Staubach et al., 2010*; *Voolstra et al., 2007*).

However, in contrast to the gene expression differences between the taxa, the sex-biased gene expression differences are a priori not caused by neutral fluctuations, as tested by the Wilcoxon significance test. This test compares the distribution of variances between the individuals of each sex for each gene to assess whether they could be part of one variance distribution or two significantly different variance distributions. The former would imply neutral processes in variance generation, while the latter must be considered to reflect selective forces to maintain their separate variance distributions.

For the somatic organs, the turnover of sex-biased gene expression is faster than the corresponding non-sex-biased gene expression (*Figure 2B*). If one assumes that most non-sex-biased gene expression evolves neutrally, this also suggests that sex-biased gene expression turnover is driven by adaptive selection processes.

Further, under a neutral divergence scenario, one would expect that different organs and different taxa show similar overall numbers of sex-biased genes and that these should be randomly recruited from the different gene expression modules. Both of these patterns are clearly not the case (see *Figures 1 and 4*). For example, with a neutral mechanism, it may be difficult to explain why DOM has such a large number of sex-biased genes in the heart, most of which belong to only one module of gene expression. Similarly, the other module-based patterns in other organs may not be readily explained by such a mechanism.

The alternative for explaining the module patterns would be that transcription factors that regulate these modules become sex-biased through neutral mechanisms, with the consequence that they drag with them a number of genes from within the respective modules to make them sex-biased. Such an influence of sex-biased transcription factors on their target genes was suggested by the analyses in *Naqvi et al., 2019*. However, this would not explain why we see a faster adaptive protein evolution among the sex-biased genes. If their sex-bias status is passively driven by neutrally evolving transcription factors, one would not expect a differential selection pressure acting on their protein sequences.

In the combination of these considerations, we favor an interpretation where positive selection may drive sex-biased gene expression, probably including transacting regulatory genes to explain the module patterns. It is generally known from selective sweep studies (*Ihle et al., 2006*; *Staubach et al., 2012*; *Teschke et al., 2008*) and amino acid substitution comparisons (*Halligan et al., 2010*)

that the mouse populations are subject to massive positive selection. The high propensity for positive selection in mice is likely due to their short generation times and large effective population sizes.

## Role of sexual selection

The sexual selection theory assumes that males and females have different interests in their reproductive strategies, which results in a continuous evolutionary conflict due to divergent trait optima expressed in adults of either sex (*Darwin, 1871*; *Hamilton, 1967*; *Mank, 2017a*; *Parsch and Ellegren, 2013*; *Price et al., 2023*). The resulting sexual dimorphisms can occur at many levels, not only as prominent major phenotypic differences, but also as very many cryptic differences, as has, e.g., been shown in a systematic survey of the phenotypes of mouse knockout strains (*van der Bijl and Mank, 2021*). Many of these cryptic differences are expected to be caused by the sex-biased expression of genes (*Grath and Parsch, 2016*; *Mank, 2017b*). Hence, there is much room for the adaptive evolution of sex-biased gene expression due to the existence of continuous sexual conflict (*Cox and Calsbeek, 2009*) or sex-specific selection, e.g., *Oliver and Monteiro, 2011*.

Sexual selection processes include intrasexual selection (competition within a sex, e.g. male-male competition) and intersexual selection (mate choice, e.g. female preference for specific male traits). There could also be sexually antagonistic selection, where a trait benefits one sex but harms the other. Either of these processes can be linked to sex-biased gene expression patterns. While our data do not allow us to distinguish between these different processes, we note that in mice the number of female-biased genes is consistently larger than male-biased genes, both across organs and across taxa. This could argue for stronger intrasexual selection processes in females, which would be in contrast to the often-assumed stronger effects in males. Mice show only little overt sexual dimorphism, although males are on average larger, in line with a stronger physical male-male competition for access to mates. However, mice are cooperative breeders, and for such breeding systems, it was suggested that there could also be intense intrasexual competition among females, e.g., for resources or higher social status (*Rosvall, 2011*; *Rubenstein and Lovette, 2009*; *Tobias et al., 2012*). Although it remains open whether the relative number of sex-biased genes is an appropriate measure for the strength of intrasexual selection, the observed larger numbers in females could suggest that strong intrasexual selection may also occur among females.

Many previous studies on sex-specific expression have focused on differences between gonads. But gonads are composed of very different cell types and tissues in males versus females. Hence, many of the sex-biased genes in gonads may simply be sex-biased because of the different cell types that generate these different tissues. Our data allow a direct comparison of evolutionary patterns in gonads versus somatic tissues, since they were taken from the same individuals. We find that the patterns are indeed different. Gonadal sex-biased genes are more conserved between the taxa, and the turnover of sex-biased gene expression is not faster than that of non-biased genes in these tissues.

Yet, gonads also produce the hormones that are required for the sex-specific differentiation of the tissues, including the gonads themselves. Hence, in populations where individuals assume intermediate reproductive tactics between males and females, it was also possible to correlate this intermediacy with corresponding patterns of sex-biased gene expression in the gonads (*Dean et al., 2017*; *Pointer et al., 2013*; *Todd et al., 2018*). In a study in fish that show individuals with female mimicry, it was also possible to find differences in brain expression patterns and modules according to reproductive tactics (*Cardoso et al., 2018*). Accordingly, there is an interplay between gonadal and sex-biased gene expression patterns, but gonadal molecular evolutionary patterns cannot be directly compared to somatic patterns.

## Intragroup variation and the SBI

Our results show that somatic sex-biased genes do not only evolve fast between lineages, but they also represent the fraction of genes that are more variable within lineages. This implies that they contribute more to individual variation than non-sex-biased genes. Interestingly, in humans, environmentally responsive genes also show higher variances than genes involved in regulating fundamental cellular processes (*Wolf et al., 2023*), and a subset of genes shows differential variability depending on which sex expresses them (*Khodursky et al., 2022*).

The SBI constitutes a cumulative index to compare the variances in sex-biased gene expression in individuals. It has some intrinsic features that make it particularly suitable to reflect the individual

variances. In contrast to measures that would focus on either male-biased or female-biased genes, it puts both into relation to each other by subtracting the respective median expressions of male sex-biased genes from those of female sex-biased genes. The use of the medians, rather than the averages, reduces the impact of genes with extremely high sex-bias because of specific molecular functions, e.g., the gene *Xist* that is required in every cell for the dosage compensation of the X-chromosomes in females. However, note that even this gene shows a rather variable expression between females (values included in Supplementary Data D1).

The distribution of overall transcriptional variances between individuals is usually done by principal component analyses (PCAs). This statistical approach tries inherently to find the components that distinguish two groups of samples. It calculates principal components (PCs) as eigenvectors of a covariance matrix, assuming linear combinations of the variables. The first principal component (PC1) captures the highest variance with the subsequent PCs reflecting additional variance components in descending order. When we apply a PCA to our data, we find that usually around 50–60% of the variance is reflected in PC1, which separates the sexes. However, all further PCs do not separate the sexes, implying that 40–50% of the variances are overlapping. Hence, the PCA supports a spectrum of overlapping variances between the sexes, but displays these in separate axes that need to be plotted separately. Further, by using linear correlation, it does not correct for outlier expression values. Hence, we consider the SBI as more suitable for visualizing variance distributions between the sexes since it captures them as a distribution of single values per individual, with correction for overexpression.

The use of the SBI as a single value also follows the general concepts of 'genomic prediction' (*Meuwissen et al., 2001*) which are based on summarizing values, reflecting the polygenic mechanism of the generation of the phenotype (*Tautz et al., 2023*). Such a cumulative index across the effects of all expressed genes in a tissue is now also being developed for understanding human genetic diseases, summarized as a polygenic risk score (*Crouch and Bodmer, 2020*; *Gibson, 2019*). The SBI represents a similar approach, but specifically for studying sex-specific differences.

The use of a subtraction rather than a ratio between each of the sex-biased medians has the advantage of implicitly retaining the information on the relative expression levels, as well as allowing a simple comparison on a linear scale. Linear scales are also usually used to compare distributions of morphological measures, especially in the context of visualizing overlapping sex-specific patterns (*Maney, 2016*). Normalized SBI values can also be used in a heatmap visualization to represent a mosaic pattern of sex-biased differences (*Joel et al., 2015*), as we have done in *Figure 6*.

The SBI distribution comparisons in *Figures 5 and 7* show for many tissues overlapping distributions, especially in humans. This implies that any individual that falls within the overlapping distributions would not be readily classifiable as belonging to either sex based on this information alone. The gonadal distributions are always very distinct and nonoverlapping, but this can be explained by the different tissue and cell types, as discussed above. Interestingly, however, we also find a distinct distribution of SBI variances for kidney and liver in DOM and MUS mouse populations. This shows that even somatic organs that do not differ in their cell types can show very distinct SBI variances. Interestingly, the two other taxa, SPR and SPI, show overlapping distributions for these organs, implying that the very distinct patterns in DOM and MUS are based on adaptive differences between the sexes. Note that it is known that the gene expression differences in these organs are directly regulated by hormone action (*Clodfelter et al., 2006*; *Laulhé et al., 2021*), but this does not explain why the strong differences exist in the first place, especially when a similarly strong differentiation is not seen in the other mouse taxa or humans.

There is a general tissue specificity of sex-biased gene expression in mammals (*Naqvi et al., 2019*; *Oliva et al., 2020*; *Rodríguez-Montes et al., 2023*; *Yang et al., 2006*), and we find a lack of correlation of SBIs between organs of the same individual. This implies that individual males or females are actually composites with respect to displaying more female or more male characters throughout their body and brain. Hence, while hormones are undoubtedly the drivers of dimorphic sex differentiation, they act via different sets of transcription factors in the different organs (*Williams and Carroll, 2009*). This allows for generating independent variances with respect to sex-biased gene expression in different cell types and different organs, which are reflected in the SBI distributions.

The generally broader overlaps in somatic SBI distribution in humans compared to mice can be ascribed to two factors. First, there are fewer sex-biased genes in humans, possibly because the relatively lower population size in humans cannot maintain so many adaptive differences as in mice (see above). Second,

the human samples are much more heterogeneous with respect to age and environmental exposure, compared to the mice that were collected at very similar ages and grew up under controlled environmental conditions. However, this implies that humans reflect the more natural conditions, while the mouse data represent a somewhat more artificial accentuation of the genetic differences. Evidently, both for the action of adaptive processes and for our own thinking about sex differences, the natural conditions are more relevant - and they seem to favor larger overlapping variances.

Our results show that most of the transcriptional underpinnings of somatic sex differences show little long-term evolutionary stability. This fast evolution is accompanied by high individual variability of sex-biased gene expression with overlapping distributions and different sex-biases in different organs. This parallels the general observation that sex-dimorphic character distributions, e.g., body height, are overlapping between the sexes. Adult individuals are therefore composed, in many somatic tissues, of a spectrum of sex characteristics that are not always captured by a simple binary classification (*Ainsworth, 2015*; *Maney, 2016*; *Sharpe et al., 2023*). This is also relevant for the consideration of sex-specific medical treatments. While average differences in disease etiology between sexes are well documented (*Mauvais-Jarvis et al., 2020*), the decision on where a given individual falls into the spectrum of maleness/femaleness differences becomes more difficult, the more the variances overlap. The fast evolution of sex-biased expression is also a warning sign that mouse models may not be suitable for developing gender-specific medicinal treatments for humans, especially in view of the fact that humans have a much less sex-biased transcriptome than mice.

## Conclusions

The present study reports the largest systematic dataset so far on the micro-evolutionary patterns of sex-biased gene expression in outbred animals. It is also the first that explores the patterns of individual variation in sex-biased gene expression, and the SBI is a new procedure to directly visualize these variance patterns in an intuitive way. Previous papers on the evolution of sex-biased gene expression in animals have mostly concentrated on gonadal organs and have suggested that the turnover of sex-biased expression is fast. Interestingly, at least at the micro-evolutionary level that we study here, the turnover of gonadal sex-biased expression of the genes is not faster than the corresponding non-sex-biased expression in the same organs. This is different for the somatic patterns, most of which evolve much faster for the sex-biased genes than for the corresponding non-sex-biased genes. This suggests that the effects of sexual selection are stronger in the somatic tissues than in the gonadal tissues. Since many genes are involved in this, this implies that evolutionary changes driven by sexual selection on somatic characters are continuously acting, even in taxa that show no strong overt sexual dimorphism. Our data also show that the turnover of many genes involved in sex-biased expression occurs within regulatory modules. We find that the very few genes that have a conserved sex-biased expression between mice and humans in more than one organ are general epigenetic regulator genes. It will therefore be interesting in the future to focus on their roles in generating the differences between the somatic sexual phenotypes in given species.

## Methods

**Key resources table**

| Reagent type (species) or resource | Designation | Source or reference | Identifiers | Additional information |
|---|---|---|---|---|
| Biological sample (*Mus musculus domesticus*) | Somatic and gonadal organs | MPI for Evolutionary Biology, Plön | FRA | Freshly dissected from animals |
| Biological sample (*Mus musculus musculus*) | Somatic and gonadal organs | MPI for Evolutionary Biology, Plön | KAZ | Freshly dissected from animals |
| Biological sample (*Mus spretus*) | Somatic and gonadal organs | MPI for Evolutionary Biology, Plön | SPR | Freshly dissected from animals |
| Biological sample (*Mus spicilegus*) | Somatic and gonadal organs | MPI for Evolutionary Biology, Plön | SPI | Freshly dissected from animals |
| Commercial assay or kit | RNeasy 96 Universal Tissue Kit | QIAGEN | Catalog no. 74881 | |

*Continued on next page*

*Continued*

| Reagent type (species) or resource | Designation | Source or reference | Identifiers | Additional information |
|---|---|---|---|---|
| Commercial assay or kit | TruSeq Stranded mRNA Kit | Illumina | 2×150 bp | Used in sequencing center Kiel |
| Software/algorithm | asymptoticMK (downloaded on 2023-2-20) | *Haller and Messer, 2017*; *Haller and Leinweber, 2017* | https://github.com/MesserLab/asymptoticMK | |
| Software/algorithm | snpEff (4.3t) | *Cingolani et al., 2012a*; *Cingolani et al., 2012b* | https://pcingola.github.io/SnpEff/ | |
| Software/algorithm | Trimmomatic (0.38) | *Bolger et al., 2014*; *Bolger et al., 2021* | https://www.usadellab.org/cms/?page=trimmomatic | |
| Software/algorithm | HISAT2 (2.2.1) | *Kim et al., 2015*; *Kim et al., 2019* | https://daehwankimlab.github.io/hisat2/download/ | |
| Software/algorithm | featureCounts (2.0.3) | *Liao et al., 2014*; *Liao et al., 2021* | https://subread.sourceforge.net/featureCounts.html | |
| Software/algorithm | WGCNA (1.71) | *Langfelder and Horvath, 2008*; *Langfelder et al., 2022* | https://edo98811.github.io/WGCNA_official_documentation/ | |

## Mouse organ samples

Gene expression data were collected from outbred individuals from two *M. musculus* subspecies, DOM (population FRA) and MUS (population KAZ), as well as two sister species, SPR and SPI. For further origin details on DOM, MUS, and SPR see Table S1 in *Harr et al., 2016*. SPI individuals were derived from catches of wild mound-building mice caught at two different locations in Western Slovakia (Bohelov: 47°54'26" N, 17°41'58" E; Sasa 48°03' N; 17°25'E - founder mice were provided by Kerstin Musolf, first described in *Musolf et al., 2015*).

All replicates for a population are biological replicates, and the number of replicates was based on power analysis considerations discussed in *Xie et al., 2020*. Nine age-matched adult females and adult males were chosen from each of the four taxa, 72 individuals are included in total in the overall analysis. As somatic organs, we included brain (whole brain), heart, liver (left medial lobe), kidney (right), and mammary gland (fourth, right). Note that the mammary glands in mice have similar sizes in both sexes before lactation and are therefore directly comparable. As gonadal organs, we chose ovary (both ovaries), oviduct (both oviducts), and uterus (right uterine horn) from females and testis (right), epididymis (right), and vas deferens (right) from males. All organs were always retrieved from the same individuals, allowing expression comparisons between organs within each individual. In total, 576 samples were included (*Figure 1—source data 2*).

## RNA sequencing and data analysis

The organs were carefully prepared and immediately frozen in liquid nitrogen. Total RNA was purified using QIAGEN RNeasy 96 Universal Tissue Kit (Catalog no. 74881), and prepared using Illumina TruSeq Stranded mRNA Kit, and sequenced using Illumina NovaSeq S4 (2×150 bp) in Kiel Sequencing Center. All procedures were performed in a standardized and parallel way to reduce experimental variance.

Raw sequencing reads were trimmed using Trimmomatic (0.38) (*Bolger et al., 2014*; *Bolger et al., 2021*). Only paired-end reads left were used for following analyses. The trimmed reads were mapped to mouse genome GRCm39 (*Martin et al., 2023*; *Mouse Genome Sequencing et al., 2002*) using HISAT2 (2.2.1) (*Kim et al., 2015*; *Kim et al., 2019*) with default parameter settings, except for '--score-min' set as 'L,0.0,–0.6', in order to compensate the sequence divergences of individuals from various taxa. Fragments mapped to the genes annotated by Ensembl (Version 104) were counted using featureCounts (2.0.3) (*Liao et al., 2014*; *Liao et al., 2021*).

## Assignment of sex-biased gene expression

Various methods have been used to identify genes with sex-biased expression, also depending on whether the focus of the study was on gonadal differences or on other organ comparisons (*Blekhman*

*et al., 2010*; *Harrison et al., 2015*; *Naqvi et al., 2019*; *Oliva et al., 2020*; *Reinius et al., 2008*; *Rodríguez-Montes et al., 2023*; *Yang et al., 2006*). Most of these methods apply some form of parametric statistics, such as a negative binomial distribution, which is necessary when only few samples are available per group. However, this is problematic in two ways. First, transcriptome data are well known for showing outliers, and second, the variances of expression can be different among different genes with similar expression level, and between males and females. Thus, these parametric methods usually lead to exaggerated false positives (*Li et al., 2022*). We used here at least nine individuals per sex, which allows us to perform nonparametric statistics, combined with setting cutoff criteria with an FDR correction. For each organ and each taxon, we normalized the fragment counts to TPM values and added one to all TPM values ('TPM +1') to avoid dealing with zeros in ratio calculations. Only genes with a median of 'TPM+1' in at least one sex >2 were kept for analysis. The sex-bias ratio for each gene is calculated as 'MEDIAN of females/MEDIAN of males'. We have explored a range of cutoffs and found that a sex-bias ratio of 1.25-fold difference of MEDIAN expression values combined with a Wilcoxon rank sum test and Benjamini-Hochberg FDR correction (using FDR<0.1 as cutoff) (*Benjamini and Hochberg, 1995*) yields the best compromise between sensitivity and specificity. This was assessed by comparing sex-randomized datasets with the actual data. In these pre-tests, we found that the inclusion of the Wilcoxon test with FDR correction was most effective in increasing the contrast between randomized scores and actual scores to at least 20-fold difference. For controls with non-biased genes, we chose a cutoff of a <1.05-fold ratio without Wilcoxon test.

For the somatic organs, we used the comparison between nine females and males each to identify genes with sex-biased expression. For the gonadal organs, we chose ovary versus testis (OvaTes), oviduct versus epididymis (OviEpi), and uterus versus vas deferens (UteVas) for the pairwise comparisons. Except for ovary versus testis, the other two pairs are not really considered to be homologous, but they serve very roughly comparable general functions.

## SBI calculation

The calculation of the SBI is based on the following principles:

1. Create a single value for each individual that summarizes its relative sex-bias.
2. Summarize across all genes except the ones on the Y-chromosome, since these cannot be expressed in females.
3. Use a statistic that reduces the influence of outlier expression (*Xie et al., 2025*).

This is achieved by using the following overall formula:

$$\text{SBI} = (\text{MEDIAN of all female-biased TPM}) - (\text{MEDIAN of all male-biased TPM})$$

Since the SBI is meant to visualize only the variances between the individuals, we normalize the values to center the distributions around 0, applying the following transformation: normalized value = ((2 × ([original value] – [smallest value in data range]))/([largest value in the data range] – [smallest value in the data range])) – 1.

The individual normalized SBI values can be plotted as density plots that reflect the spectrum of variances. To visualize these as a continuous distribution, we use the kernel density estimation with a Gaussian kernel to create a smoothed density plot (implemented in the R function ggplot2::geom_density with the parameters 'adjust = 3, alpha = 0.5').

In this display, the females should show a range of positive values representing their individual variation, and the males should show an analogous range of negative values. The corresponding density functions can range from strong overlap to strong separation.

## Lists of all steps of the full sex-bias analysis procedures
### Identification of sex-biased genes

1. Normalize the fragment counts to TPM values.
2. Add 1 to all TPM values to avoid dealing with zeros in ratio calculations.
3. Choose equal numbers of males and females.
4. Each calculation is separate for each organ and each species.
5. Minimal expression cutoff in at least one sex for a given organ in a given taxon: median >2 (=cutoff at 1 TPM plus 1 added).

6. Calculate MEDIAN expression separately across all females and all males.
7. Calculate the nonparametric Wilcoxon rank sum test for females versus males for each gene.
8. Calculate the sex-bias ratio for each gene: MEDIAN females/MEDIAN males.
9. Set FEMALE BIAS at >1.25.
10. Set MALE BIAS at <0.8.
11. Set NEUTRAL at >1/1.05 to <1.05.
12. Generate the subsets of genes with FEMALE BIAS or MALE BIAS.
13. For the combined subsets, calculate correction for multiple testing (FDR according to Benjamini-Hochberg) for the p-values from the Wilcoxon rank sum test.
14. Use FDR<0.1 as cutoff for significantly sex-biased genes.

## Calculation of the SBI

1. Calculate for each individual the MEDIAN of the expression values (TPM counts) for FEMALE BIAS genes and MALE BIAS genes.
2. SBI = MEDIAN of FEMALE BIAS – MEDIAN of MALE BIAS.
3. Center data around 0: [value] = ((2 × ([original value] – [smallest value in data range]))/([largest value in the data range] – [smallest value in the data range])) – 1.
4. Generate SBI values as histograms across all females and all males.
5. Use the smoothing function implemented in ggplot2: geom_density to generate continuous distributions.

## Calculation of sex-specific variances

1. Calculate IQR/median for each gene and each sex in each organ and each taxon.
2. Plot the distributions of IQR/median for FEMALE BIAS, MALE BIAS, NEUTRAL, and RECIP-ROCAL genes.

## Calculation of regulatory turnover of all genes between taxa

Note: this procedure is based on the same overall statistic as it is done for identifying sex-biased genes; a changed regulation is defined as a at least 1.25-fold difference in TPM values, plus Wilcoxon test with FDR<0.1.

1. Female swaps: replace male data from a given taxon with female data from the compared taxon.
2. Male swaps: replace female data from a given taxon with male data from the compared taxon.
3. Contrasts made for (1) and (2): DOM females versus MUS females; MUS females versus SPR females; SPI females versus SPR females; and DOM males versus MUS males; MUS males versus SPR males; SPI males versus SPR males.
4. Calculate the percentage of changed genes with respect to the total number of expressed genes in the given organ.
5. Repeat this step for 1000 random draws of 1000 genes each and calculate average and standard deviation (SD).

### MK test

The asymptotic MK test (*Messer and Petrov, 2013*) was performed based on the single nucleotide variants (SNVs) of wild mouse samples derived from the whole-genome sequencing data of eight mice of each of the three taxa: DOM, MUS, and SPR (downloaded from https://wwwuser.gwdguser.de/~evolbio/evolgen/wildmouse/vcf/; *Harr et al., 2016*). The SNVs were annotated as synonymous and nonsynonymous using snpEff (4.3t) (*Cingolani et al., 2012a*; *Cingolani et al., 2012b*). For each SNV in each taxon, its type, polymorphic or fixed, was determined; and the allele frequency was calculated if it is polymorphic. A fixed SNV in *M. spretus* was considered to be the ancestral allele, and only such sites were used for the analysis. When such a site showed a fixed difference in either DOM or MUS, it was classified as derived substitution. Similarly, when the site was polymorphic, the derived allele was determined according to the fixed SNV in *M. spretus*. Variants in DOM and MUS were analyzed independently. Only SNVs on the autosomes were included for the analysis. The numbers of synonymous and nonsynonymous substitution sites and the numbers of synonymous and nonsynonymous polymorphic sites at each derived allele frequency were used as input for the adapted R script

'asymptoticMK_local.R' downloaded from https://github.com/MesserLab/asymptoticMK (*Haller and Leinweber, 2017*).

The asymptotic MK test was performed on eight sets of genes: DOM female-biased genes, DOM male-biased genes, DOM non-biased genes, DOM non-biased genes of which the orthologs are sex-biased in MUS as control, MUS female-biased genes, MUS non-biased genes, MUS male-biased genes, and MUS non-biased genes of which the orthologs are sex-biased in DOM as control.

To obtain the distribution of alpha in each MK test, we used 1000 rounds of resampling via bootstrapping from variants. The results are included in *Figure 3—source data 2*.

## WGCNA

In order to assess co-regulatory modules among the sex-biased genes, we used the WGCNA (1.71) approach (*Langfelder and Horvath, 2008*; *Langfelder et al., 2022*) to generate module assignments. According to its manual, our sample size, nine, is too low to generate biologically meaningful gene networks. We made, therefore, use of data from a parallel study (*Xie et al., 2025*) for which we obtained 39 additional DOM female samples for each of the somatic organs in the present analysis (brain, heart, kidney, liver, and mammary gland). Note that all the samples were obtained in parallel at the same time by the same people and with the same procedure. The total sample size, 48, is sufficient for a reliable WGCNA. Following the standard procedure, we first removed the outlier samples: none for brain and mammary, and one for each of heart (H35232), kidney (H35331), and liver (H35302). Then, we chose the parameter 'power' with the help of its 'pickSoftThreshold' function as 4 for brain, 8 for heart, kidney, and mammary, and 6 for liver. Finally, we ran its main function 'blockwiseModules' to get module assignments of genes in each organ with parameters as "corType = 'bicor', maxPOutliers = 0.1, networkType = 'signed hybrid', TOMType = 'signed'". The module assignments for all genes are included in the Supplementary Data D1.

## Data from humans

The human bulk RNA-Seq data were retrieved from the GTEx project (*Lonsdale et al., 2013*) as TPM files for each organ. To make them best comparable with the mouse data, we chose nine females and nine males for each organ, using the following criteria: maximum age 49 years, death reason category not 0. The full sample list is provided in *Figure 7—source data 2*. Note that in contrast to the mouse data, the different organ data come from different sets of individuals.

The human single-nucleus RNA-Seq data were retrieved from the SEA-AD project (*Gabitto et al., 2024*). 'h5ad' format files, including the normalized read counts using a log transformation of pseudo-counts per 10,000 reads, ln(CPTT+1), were downloaded from CZ CELLxGENE (*Abdulla et al., 2023*; *Megill et al., 2021*). Cells in two brain regions were included: MTG and DLPFC. Considering that it is a study of Alzheimer's disease, we specifically chose 'control' samples from individuals without dementia, and with race as 'white' for analysis. We only used the data sequenced by the '10x3' v3' assay. These criteria lead to seven females and seven males for MTG and six of them each for DLPFC. Only cell types ('subclass') having at least 100 cells for each of all individuals were kept. The full list of cell types and individuals included is provided in *Table 1—source data 1*. The expression level of a gene in a cell type of a brain region from an individual was calculated as the mean of the pseudo-counts per 10,000 reads of all the cells belonging to the cell type, mean(CPTT+1).

## Acknowledgements

We thank Christine Pfeifle and Heike Harre for animal care taking and organ preparations, Michaela Schwarz for RNA purification, Derk Wachsmuth and Kristian Ullrich for IT support, and Wenyu Zhang, Luisa Pallares, and Rui Oliveira for discussion and suggestions. The study was funded by institutional resources of the MPG.

## Additional information

### Competing interests

Diethard Tautz: Reviewing editor, eLife. The other authors declare that no competing interests exist.

## Funding

| Funder | Grant reference number | Author |
|---|---|---|
| Max-Planck-Gesellschaft zur Förderung der Wissenschaften | | Chen Xie Sven Künzel Diethard Tautz |

The funders had no role in study design, data collection and interpretation, or the decision to submit the work for publication. Open access funding provided by Max Planck Society.

## Author contributions

Chen Xie, Conceptualization, Resources, Data curation, Software, Formal analysis, Validation, Investigation, Visualization, Methodology, Writing – review and editing; Sven Künzel, Resources, Investigation, Methodology; Diethard Tautz, Conceptualization, Formal analysis, Supervision, Funding acquisition, Validation, Visualization, Writing – original draft, Project administration, Writing – review and editing

## Author ORCIDs

Chen Xie ⓘ https://orcid.org/0000-0002-6183-7301
Sven Künzel ⓘ http://orcid.org/0000-0003-4992-5963
Diethard Tautz ⓘ https://orcid.org/0000-0002-0460-5344

## Ethics

Maintenance and handling of the animals were conducted in accordance with German Animal Welfare Act and FELASA guidelines. The project was approved with the number 1158 by the Animal Welfare Officers of the University of Kiel according to the German Animal Welfare Act §4 "Killing animals and organ withdrawal for scientific purpose". Permits for keeping mice were obtained from the veterinary office "Veterinäramt Kreis Plön" under permit number: PLÖ;-000 4697 (08.04.2014).

Reviewer #4 (Public review): https://doi.org/10.7554/eLife.99602.4.sa1
Reviewer #5 (Public review): https://doi.org/10.7554/eLife.99602.4.sa2
Author response https://doi.org/10.7554/eLife.99602.4.sa3

# Additional files

## Supplementary files
MDAR checklist

## Data availability

Sequencing data have been deposited in ENA under BioProject accession number PRJEB50011. All data generated or analysed during this study are included in the manuscript and supporting files; source data files have been provided for the figures. The tables with TPM values and statistical analysis for all genes and organs are available as supplemental data folders D1 - D3 at the Edmond repository of the MPG: https://doi.org/10.17617/3.IMARWT. The code for sex-biased gene detection and analysis is provided on GitHub (copy archived at *Xie, 2025*).

The following datasets were generated:

| Author(s) | Year | Dataset title | Dataset URL | Database and Identifier |
|---|---|---|---|---|
| Xie C, Tautz D | 2024 | RNA-Seq of samples from four wild mouse taxa | https://www.ebi.ac.uk/ena/browser/view/PRJEB50011 | European Nucleotide Archive, PRJEB50011 |
| Xie C, Tautz D | 2025 | Transcriptome data on sex-biased gene expression in mouse and human | https://doi.org/10.17617/3.IMARWT | Edmond, 10.17617/3.IMARWT |

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

## Appendix 1

### Supplementary data D1 (*Xie and Tautz, 2025*)

Mouse transcriptome data for all taxa and organs in the study

The D1 folder includes eight Excel files, one for each organ comparison (brain, heart, kidney, liver, mammary, OvaTes, OviEpi, UteVas), each with three tables. Sheet 1 (<organ > _all) includes the list of all 55,416 genes with metadata, relevant statistical columns, and TPM+1 data for each individual. Sheet 2 (summary stats) includes the same gene list with sex-bias assignment and conservation statistics. Sheet 3 (<organ > sex-biased) includes the subset of genes that are sex-biased in any of the four taxa.

### Supplementary data D2 (*Xie and Tautz, 2025*)

Swap data tables for the mouse transcriptome data

Based on the mouse transcriptome data for all taxa and organs from D1, but used for the swap study to estimate the effects of neutral evolution. The D2 folder includes eight Excel files, one for each organ comparison (brain, heart, kidney, liver, mammary, OvaTes, OviEpi, UteVas), each with three tables. Sheet 1 (<organ > sex-biased) includes the list of all 55,416 genes with statistical data from the D1 data tables and analysis columns in CW to DC with excel formulas. Tab 2 (<organ > female swaps) includes the same gene list but swapped females between taxa and analysis columns in CW to DC with Excel formulas. Sheet3 (<organ > male swaps) includes the same gene list but swapped males between taxa and analysis columns in CW to DC with Excel formulas.

### Supplementary data D3 (*Xie and Tautz, 2025*)

Human transcriptome data from GTEx used for the study

The D3 folder includes 10 Excel files, one for each organ comparison that was used for *Figure 7*, each with two tables. Sheet 1 (<organ > _all) includes the list of all 56,200 genes with statistic columns and all individuals in the study. Sheet 2 (<organ > _SB genes) lists the subset of sex-biased genes.

