## [Editor Report · eLife Assessment]

This study presents data on sex differences in gene expression across organs of four mice taxa. The authors have generated a unique and **convincing** dataset that fills a gap left by previous studies. They claim that sex-biased expression in the soma can overlap between genetic males and females, and that the relevant patterns both turn over quickly over short evolutionary times and do so faster in somatic than gonadal tissues. These conclusions could largely have been predicted by extrapolating from previous findings in the field, but nevertheless demonstrating them directly is a **fundamental** advance.

[Editorial note: The work was originally assessed by colleagues who are active in the field of evolution of sex differences or in areas adjacent to this field (see initial assessment at https://doi.org/10.7554/eLife.99602.2). The appeals process involved consultation with experts working in other areas of evolutionary biology. The above assessment synthesises the opinions of both sets of reviewers.]

---

## [Referee Report · Reviewer #4 (Public review)]

The paper by Xie et al. investigates the micro-evolutionary dynamics of sex-biased gene expression across somatic and gonadal tissues in four mouse taxa, with comparative analyses in humans. The study introduces a new metric, the Sex-Bias Index (SBI), to quantify individual-level variation in sex-biased gene expression, and explores the evolutionary turnover, variance, and adaptive evolution of these genes.

These strengths of the paper are not in dispute:

Novelty: The study is among the first to systematically analyze sex-biased gene expression at a micro-evolutionary scale in outbred animals, using closely related mouse taxa. This contrasts with most previous work, which focused on macro-evolutionary comparisons between distant species.

Controlled Sampling: The use of age-matched, outbred individuals raised under standardized conditions minimizes environmental confounders, allowing for robust within- and between-taxon comparisons.

Somatic vs. Gonadal Focus: Unlike many earlier studies that emphasized gonadal tissues, this work provides a detailed analysis of somatic organs, revealing rapid evolutionary turnover and mosaicism in sex-biased gene expression.

Sex-Bias Index (SBI): The SBI offers a cumulative, individual-level measure of sex-biased gene expression, facilitating visualization of variance and overlap between sexes within tissues. While one can argue about whether a new metric is necessary (as the authors argue), the combination of fold-change cutoffs, non-parametric Wilcoxon tests, and FDR correction reduces false positives, addressing concerns raised in the field about inflated detection of sex-biased genes.

Evolutionary implications: The study demonstrates that sex-biased gene expression in somatic tissues evolves more rapidly than in gonads, and that this turnover is often accompanied by signatures of adaptive protein evolution. The lack of correlation in SBI across tissues within individuals supports a mosaic model of sex-biased gene expression, challenging binary models of sexual differentiation.

The weaknesses are already listed by previous rounds of review but I will add one more: in an attempt to be comprehensive, the writing is quite dry and the main conclusions sort of get hidden within the less important observations.

Since the debate is mostly about what words to use to describe the importance and the strength of evidence, I thought it would be useful to directly compare this study to other studies that address the same topic:

Naqvi et al. Science 2019 (David Page lab): Conservation, acquisition, and functional impact of sex-biased gene expression in mammals

Oliva et al. Science 2020 (Stranger lab): The impact of sex on gene expression across human tissues

Rodríguez-Montes et al. Science 2023 (Kaessman, Cardoso-Moreira labs)

Let's start with the fact that all three peer studies have had a major impact. Second, although Naqvi et al. (2019) and Oliva et al. (2020) provided foundational cross-species and cross-tissue analyses of sex-biased gene expression, but did not address micro-evolutionary turnover or individual-level variance. Third, Rodríguez-Montes et al. (2023) focused on developmental and evolutionary patterns of sex-biased expression, but at a broader phylogenetic scale and without the individual-level or module-based analyses presented here. None of the peer studies addressed the possibility of mosaicism within individuals, none of them addressed the relations between expression bias and adaptive evolution. So the comparison is really a bit of an apples to oranges comparison: the peer studies are about patterns in deep phylogeny, whereas the present study is an amazing (to me) analysis of inter-individual mosaicism, which is at the heart of this kind of variation, which would totally be missed or worse misinterpreted in deep phylogenetic analyses. Having said that, in my subjective opinion, all three related papers are better written than the present one, but to me there is no question this belongs in the same pedestal as all of them.

---

## [Referee Report · Reviewer #5 (Public review)]

Xie et al. present a data set of impressive size to study changes in sex-biased gene expression. A clear strength that sets the study apart from previous work is the use of age-matched outbred individuals raised in the same environment, which minimizes non-genetic variance, and the comparison of closely related taxa. Also in contrast to many previous studies, while gonads, which have often been the focus of sex-biased gene expression studies, are not ignored, multiple gonadal tissues are being compared to an array of somatic tissues. The study design therefore can offer a particularly rich and nuanced view of how sex differences change across tissues and over short evolutionary times.

I liked the idea of summarizing over the mean expression of gene sets, instead of just using numbers of DEGs for comparisons, even though the introduction of the term "Sex-Biased Index (SBI)" seems somewhat of an overkill. The summary analyses are definitely useful to visualize variability in sex-biased gene expression programs. The authors find that the expression patterns of sex-biased genes change faster than those of non-sex-biased genes - but only in somatic tissues. They also provide some evidence that this correlates with higher rates of potentially adaptive coding sequence changes in the taxa where expression is sex-biased, with the proviso that a stronger modeling framework would have made these inferences more robust.

I was most surprised by the finding that the fast change in expression patterns is linked to different gene expression modules becoming sex-biased in the different taxa studied. This is in my eyes a remarkable observation that could not have been predicted from previous knowledge.

The use of human GTEx and patient scRNA-seq data is a nice addition, although there are known confounding issues with these resources, given that these are not random samples and environmental conditions are uncontrolled. Nevertheless, as the human data echo the trends seen with the much more rigorous mouse data set, I do not have principal objections to this addition. Furthermore, the human data do allow the authors to conclude that only very few genes with sex-biased expression are shared in the soma of mice and humans.

In summary, I believe that this contribution has the potential to fundamentally change how we see sex-biased gene expression differences in vertebrates, given that the author's conclusions are grounded in a data set of compelling quality and size.

---

## [Author Response]

The following is the authors’ response to the previous reviews

**Reviewer #2 (Public review):**
Summary:The manuscript by Xie and colleagues presents transcriptomic experiments that measure gene expression in eight different tissues taken from adult female and male mice from four species. These data are used to make inferences regarding the evolution of sex-biased gene expression across these taxa.Strengths:The experimental methods and data analysis appear appropriate. The authors promote their study as unprecedented in its size and technical precision.

We do not understand the statement "the authors promote" as if there was a doubt about this. If there is a doubt, we welcome to see it specified.

Weaknesses:The manuscript does not present a clear set of novel evolutionary conclusions. The major findings recapitulate many previous comparative transcriptomics studies - gene expression variation is prevalent between individuals, sexes, and species; and genes with sex-biased expression evolve more rapidly than genes with unbiased expression - but it is not clear how the study extends our understanding of gene expression or its evolution.

There have been no "previous comparative transcriptomics studies" at a micro- evolutionary scale in animals, hence, we do not "replicate" these. And our contrast between somatic and gonadal patterns reveals insights that have not been recognized before, namely that gonadal sex-specific expression turnover is actually not faster that the corresponding non-sex-specific truover. We have now further clarified this distinction throughout the text and have also adapted the title of the paper accordingly.

We agree with the overall statement that "gene expression variation is prevalent between individuals, sexes, and species" but the aspect of "sex-biased gene expression between individuals" has not been systematically analysed before in such a context.

Concerning the statement that "genes with sex-biased expression evolve more rapidly than genes with unbiased expression", we note that this is mostly derived from gonadal data and that there is no study that has quantified this so far at a population level and between subspecies in comparison to somatic data.

Our results show further that previous assumptions of a substantial set of genes with sex- biased expression conserved between mice and humans are due to underestimating the convergence issues when there is an extremly fast turnover of sex-biased gene expression. This has a major implication for using mice as a model for gender-speficic medicine questions in humans.

Many gene expression differences between individual animals are selectively neutral, because these differences in mRNA concentration are buffered at the level of translation, or differences in protein abundance have no effect on cellular or organismal function. The hypothesis that sex-biased genes are enriched for selectively neutral expression differences is supported by the excess of inter-individual expression variance and inter-specific expression differences in sex-biased genes.

This statement repeats a statement from the first round of reviews. We had added new data and extensive discussion on this topic. We do not understand why this has not been taken into account. In fact, a major strength of our paper is that it shows that most sex- biased gene expression differences are not neutral!

There are two major issues here: to identify sex-biased gene expression in the first place, we (and all other papers in the field) use the neutral model as null-hypothesis. Genes that are not compatible with this null-hypothesis are considered sex-biased. In contrast to most previous papers, we have the possibility to take into account the variances between individuals to add an additional significance test. Hence, we can apply a much more rigorous two-step process: first a ratio-cutoff plus a Wilcoxon rank sum test with correction for multiple testing to identify significant deviations from the null-hypothesis. We have added some additional statements in the Results and Discussion sections to emphasize this.Second, by focusing on the genes that are not following a neutral model, the variance and divergences data support the action of selection, rather than neutral drift.

A higher rate of adaptive coding evolution is inferred among sex-biased genes as a group, but it is not clear whether this signal is driven by many sex-biased genes experiencing a little positive selection, or a few sex-biased genes experiencing a lot of positive selection, so the relationship between expression and protein-coding evolution remains unclear.

Again, there are two major issues here. First, the distribution of alpha-values shown in Figure 3B are rather homogeneous, i.e. there is not support for a scenario that the average is driven by only a few genes.

Second, it seems that the referee wants to see an analysis where dn/ds ratios are broken down for every single gene. This has been done in previous papers, but it is now understood that this procedure is fraught with error because of the demographic contingencies inherent to natural populations that can yield wrong results for individual loci. We have added some statements to the text to clarify this further.

It is likely that only a subset of the gene expression differences detected here will have phenotypic effects relevant for fitness or medicine, but without some idea of how many or which genes comprise this subset, it is difficult to interpret the results in this context.

It is the basic underlying assumption for the whole research field that significantly sex- biased genes are phenotypically relevant for fitness, since they would otherwise not be sex- biased in the first place.

Throughout the paper the concepts of sexual selection and sexually antagonistic selection are conflated; while both modes of selection can drive the evolution of sexually dimorphic gene expression, the conditions promoting and consequence of both kinds of selection are different, and the manuscript is not clear about the significance of the results for either mode of selection.

We had explained in our previous response that our data collection was not designed to distinguish between these two processes. But given that the issue is being brought up again, we have now added some discussion on this issue.

The manuscript's conclusion that "most of the genetic underpinnings of sex-differences show no long-term evolutionary stability" is not supported by the data, which measured gene expression phenotypes but did not investigate the underlying genetic variation causing these differences between individuals, sexes, or species.

We agree that - under a strict definition - our use of the term "genetic underpinning" in this conclusion sentence can be criticized. The most correct term would be "transcriptional underpinnings", but of course, given that it is the current practice of the whole field to assume that "transcriptional" is part of the overall genetics, we do not consider our initial statement as incorrect. Still, we have changed the term accordingly.

Furthermore, most of the gene expression differences are observed between sex-specific organs such as testes and ovaries, which are downstream of the sex-determination pathway that is conserved in these four mouse species, so these conclusions are limited to gene expression phenotypes in somatic organs shared by the sexes.

Yes - correct. But the whole focus of the paper is on somatic expression, i.e. organs that share the same cell compositions. Of course, the comparison between gonadal organs is conflated by being composed of different cell types. We have extended the discussion of this point.

The differences between sex-biased expression in mice and humans are attributed to differences in the two species effective population sizes; but the human samples have significantly more environmental variation than the mouse samples taken from age-matched animals reared in controlled conditions, which could also explain the observed pattern.

These are indeed the two alternative explanations that we had discussed (last paragraph of the discussion section, now the penultimate paragraph).

The smoothed density plots in Figure 5 are confusing and misleading. Examining the individual SBI values in Table S9 reveals that all of the female and male SBI values for each species and organ are non-overlapping, with the exception of the heart in domesticus and mammary gland in musculus, where one male and one female individual fall within the range of the other sex. The smoothed plots therefore exaggerate the overlap between the sexes;

Smoothing across discrete values is an entirely standard procedure for continuous variables. It allows to visualize the inherent data trends that cannot easily be glanced from simple inspection of the actual values. This is a mathematical procedure, not an "exaggeration". We used the same smoothening procedure for all the comparisons, and it is clear that the distributions between females and males of the sex organs and a few somatic organs are well separated (non-overlapping), which serves as a control.

in particular, the extreme variation shown in the SBI in the mammary glands in spretus females and spicilegus males is hard to understand given the normalized values in Table S3. The R code used to generate the smoothed plots is not included in the Github repository, so it is not possible to independently recreate those plots from the underlying data.

We apologize that there was indeed an error in the Figure - the columns for SPR and SPI were accidentally interchanged. We have corrected this figure. Generally, the smoothened patterns we show are easily verified by looking up the respective primary values. We apologize that the code lines for the plots were accidentally omitted. We have used a standard function from ggplot2: geom_density, with "adjust=3, alpha=0.5" for all plots and included this description in the Methods. We have now added this to the R code in the GitHub repository.

The correlations provided in Table S9 are confusing - most of the reported correlations are 1.0, which are not recovered when using the SBI values in Table S9, and which does not support the manuscript's assertion that sex-biased gene expression can vary between organs within an individual. Indeed, using the SBI values in Table S9, many correlations across organs are negative, which is expected given the description of the result in the text.

There is a misunderstanding here. The tables do not report correlations, but only p-values for correlations, the raw ones and the ones after corrections for multiple testing. P = 1.0 means no significant correlation. We have adjusted the caption of this table to clarify this further.

**Reviewer #3 (Public review):**
This manuscript reports interesting data on sex differences in expression across several somatic and reproductive tissues among 4 mice species or subspecies. The focus is on sex- biased expression in the somatic tissues, where the authors report high rates of turnover such that the majority of sex-biased genes are only sex-biased in one or two taxa. The authors show sex-biased genes have higher expression variance than unbiased genes but also provide some evidence that sex-bias is likely to evolve from genes with higher expression variance. The authors find that sex-biased genes (both female- and male-biased) experience more adaptive evolution (i.e., higher alpha values) than unbiased genes. The authors develop a summary statistic (Sex-Bias Index, SBI) of each individual's degree of sex- bias for a given tissue. They show that the distribution of SBI values often overlap considerably for somatic (but not reproductive) tissues and that SBI values are not correlated across tissues, which they interpret as indicating an individual can be relatively "male-like" in one tissue and relatively "female-like" in another tissue.

This is a good summary of the data, but we are puzzled that it does not include the completely new module analysis and the finding of extremely fast evolution of sex-biased somatic gene expression compared to the gonadal one.

Though the data are interesting, there are some disappointing aspects to how the authors have chosen to present the work. For example, their criteria for sex-bias requires an expression ratio of one sex to the other of 1.25. A reasonably large fraction of the "sex- biased genes" have ratios just beyond this cut-off (Fig. S1). A gene which has a ratio of 1.27 in taxa 1 can be declared as "sex-biased" but which has a ratio of 1.23 in taxa 2 will not be declared as "sex-biased". It is impossible to know from how the data are presented in the main text the extent to which the supposed very high turnover represents substantial changes in dimorphic expression. A simple plot of the expression sex ratio of taxa 1 vs taxa 2 would be illuminating but the authors declined this suggestion.

Choosing a cutoff is the standard practice when dealing with continuously distributed data. As we have pointed out, we looked at various cutoff options and decided to use the present one, based on the observed data distributions. Note that some studies have used even lower ones (e.g. 1.1). To visualize the data distribution, we had provided the overall distribution of ratios, because one would have to look at many more plots otherwise. But we have now also added individual plots as Figure 1, Figure supplement 2, as requested. They confirm what is also evident from the overall plots, namely that most ratio changes are larger than the incremental values suggested by the reviewer. Note that the original data are of course also available for inspection.

I was particularly intrigued by the authors' inference of the proportion of adaptive substitutions ("alpha") in different gene sets. The show alpha is higher for sex-biased than unbiased genes and nicely shows that the genes that are unbiased in focal taxa but sex- biased in the sister taxa also have low alpha. It would be even stronger that sex-bias is associated with adaptive evolution to estimate alpha for only those genes that are sex- biased in the focal taxa but not in the sister taxa (the current version estimates alpha on all sex-biased genes within the focal taxa, both those that are sex-biased and those that are unbiased in the sister taxa).

We have added the respective values in the results section, but since fewer genes are involved, they are less comparable to the other sets of genes. Still, the tendencies remain.

The author's Sex Bias Index is measured in an individual sample as: SBI = median(TPM of female-biased genes) - median(TPM of male-biased genes). This index has some strange properties when one works through some toy examples (though any summary statistic will have limitations). The authors do little to jointly discuss the merits and limitations of this metric. It would have been interesting to examine their two key points (degree of overlapping distributions between sexes and correlation across tissues) using other individual measures of sex-bias.

We had responded to this comment before (including the explanation that it has no strange properties when one applies the normalization that is now implemented) and we have added a whole section devoted to the discussion of the merits of the SBI. We do not know which other "individual measures of sex-bias" this should be compared to. Still, we have now added a paragraph in the discussion about using PCA as an alternative to show that this would result in similar conclusions, but is technically less suitable for this purpose.

Figure 5 shows symmetric gaussian-looking distributions of SBI but it makes me wonder to what extent this is the magic of model fitting software as there are only 9 data points underlying each distribution. Whereas Figure 5 shows many broadly overlapping distributions for SBI, Figure 6 seems to suggest the sexes are quite well separated for SBI (e.g., brain in MUS, heart in DOM).

We use a standard fitting function in R (see above), which tries to fit a normalized distribution, but this function can also add an additional peak when the data are too heterogeneous (e.g. Mammary in Figure 7).

Fig. S1 should be shown as the log(F/M) ratio so it is easier to see the symmetry, or lack thereof, of female and male-biased genes.

The log will work differently for values <1, compared to values >1 when used in a single plot. We have now generated combined plots with symmetric values to allow a better comparability.

It is important to note that for the variance analysis that IQR/median was calculated for each gene within each sex for each tissue. This is a key piece of information that should be in the methods or legend of the main figure (not buried in Supplemental Table 17).

We have now moved these descriptions into the Methods section.